# Systemic inflammation impairs microglial Aβ clearance through NLRP3 inflammasome

Dario Tejera[1,2,†], Dilek Mercan[1,†], Juan M Sanchez-Caro[1], Mor Hanan[3], David Greenberg[3], Hermona Soreq[3] (iD), Eicke Latz[2,4,5], Douglas Golenbock[4] & Michael T Heneka[1,2,4,*] (iD)

## Abstract

**Alzheimer's disease is the most prevalent type of dementia and is caused by the deposition of extracellular amyloid-beta and abnormal tau phosphorylation. Neuroinflammation has emerged as an additional pathological component. Microglia, representing the brain's major innate immune cells, play an important role during Alzheimer's. Once activated, microglia show changes in their morphology, characterized by a retraction of cell processes. Systemic inflammation is known to increase the risk for cognitive decline in human neurogenerative diseases including Alzheimer's. Here, we assess for the first time microglial changes upon a peripheral immune challenge in the context of aging and Alzheimer's *in vivo*, using *2-photon* laser scanning microscopy. Microglia were monitored at 2 and 10 days post-challenge by lipopolysaccharide. Microglia exhibited a reduction in the number of branches and the area covered at 2 days, a phenomenon that resolved at 10 days. Systemic inflammation reduced microglial clearance of amyloid-beta in APP/PS1 mice. NLRP3 inflammasome knockout blocked many of the observed microglial changes upon lipopolysaccharide, including alterations in microglial morphology and amyloid pathology. NLRP3 inhibition may thus represent a novel therapeutic target that may protect the brain from toxic peripheral inflammation during systemic infection.**

**Keywords** 2-photon; Alzheimer's; amyloid-beta; microglia; neuroinflammation

**Subject Categories** Immunology; Neuroscience

**The EMBO Journal (2019) 38: e101064**

## Introduction

Traditionally, the brain has been conceived as an immune-privileged organ. However, it is now widely accepted that several factors including obesity, acute injuries, aging, and neurodegenerative disease can trigger a sustained immune response in the central nervous system (CNS) leading to neuronal dysfunction and demise by microglia activation and release of neuroinflammatory mediators (Lucin & Wyss-Coray, 2009; Villeda *et al*, 2014; Heneka *et al*, 2015). Alzheimer's disease (AD) is the most prevalent type of dementia affecting approximately 45 million people worldwide. Pathologically, AD is characterized by the deposition of amyloid-β (Aβ), the formation of neurofibrillary tangles and neuroinflammation (Heneka *et al*, 2014). The hypothesis that innate immune activation contributes to AD pathogenesis has recently been supported by genome-wide association studies which have identified several immune-related gene variants, including *Trem2* (Guerreiro *et al*, 2013) and *Cd33* (Bradshaw *et al*, 2013), that modify the risk of developing AD. Under physiological conditions, microglia, the immune resident cells of the CNS, exhibit highly ramified and motile cell processes that allow continued surveillance of their environment for tissue damage, cell debris, or pathogens. Once microglia sense signals indicating such challenges, they react in order to maintain cerebral homeostasis (Davalos *et al*, 2005; Tremblay *et al*, 2010). During aging and neurodegeneration, microglia acquire an activated phenotype, morphologically characterized by a reduction in branch number accompanied by an increase in cell soma volume. Functionally, microglial activation is defined by the release of pro-inflammatory cytokines such as IL-1β, TNF-α, and IL-6 (Biber *et al*, 2007; Wyss-Coray & Rogers, 2012; Tejera & Heneka, 2016). The initiation of the inflammatory response by microglia involves the multiprotein complexes termed "inflammasomes". Comprising a cytosolic multiprotein platform, the inflammasome enables the activation of pro-inflammatory caspases, mainly caspase-1. NACHT-, LRR-, and pyrin (PYD)-domain-containing protein 3 (NLRP3) inflammasome is the best-characterized and most widely implicated regulator of IL-1β and IL-18 (Lu *et al*, 2014; Walsh *et al*, 2014). In AD, microglia are activated upon deposition of fibrillar Aβ presumably as an attempt to remove Aβ aggregates. It has been shown that this process is highly dependent on the NLRP3 inflammasome (Halle *et al*, 2008; Heneka *et al*, 2013). Because Aβ deposition is an early

1  Department of Neurodegenerative Disease and Geriatric Psychiatry, University Hospitals Bonn, Bonn, Germany
2  German Center for Neurodegenerative Diseases (DZNE), Bonn, Germany
3  Department of Biological Chemistry, The Alexander Silberman Institute of Life Sciences, The Hebrew University of Jerusalem, Jerusalem, Israel
4  Department of Infectious Diseases and Immunology, University of Massachusetts Medical School, Worcester, MA, USA
5  Institute of Innate Immunity, University Hospitals Bonn, Bonn, Germany
   *Corresponding author. Tel: +49 228 28713091; Fax: +49 228 28713166; E-mail: michael.heneka@ukbonn.de
   †These authors contributed equally to this work

event, preceding the development of mnestic and cognitive deficits by years if not decades (Jack *et al*, 2013), it is likely that microglial activation influences the pathogenesis of AD during this clinically silent period. It is therefore important to identify exogenous and endogenous factors that influence these microglial responses and thereby the pathogenesis of AD. There is considerable evidence suggesting that systemic inflammation triggers a neuroinflammatory response, characterized by sustained microglial activation with deleterious consequences for learning and memory in rodent models (Semmler *et al*, 2005, 2007; Weberpals *et al*, 2009) and in human patients (Qin *et al*, 2007; Semmler *et al*, 2008, 2013; Iwashyna *et al*, 2010; Gyoneva *et al*, 2014; Widmann & Heneka, 2014). Additionally, it has been proposed that systemic inflammation could influence the pathogenesis of different neurodegenerative diseases such as Alzheimer's disease (AD), Parkinson's disease, and multiple sclerosis (Cunningham *et al*, 2005; Qin *et al*, 2007; Cardoso *et al*, 2015). While microglia-driven neuroinflammation has been identified as a key process during systemic inflammation, aging, and neurodegenerative diseases, its dynamics *in vivo* and mechanism remain poorly understood. Here, using *in vivo* two-photon laser scanning microscopy (2PLSM), we describe the effects of systemic inflammation and aging on microglia activation. Moreover, we determine how systemic inflammation alters Aβ pathology by negatively regulating microglial clearance capacity. On a mechanistic level, we identify the NLRP3 inflammasome-signaling pathway as a key mediator of detrimental microglial effects during aging and systemic inflammation.

# Results

## Systemic inflammation affects microglia in an age-dependent manner

Previous reports demonstrated that the peripheral administration of a single dose of lipopolysaccharide (LPS) ranging from systemic inflammation (0.5–1 mg/kg) to septic shock dosages (5–10 mg/kg) causes an immune response in the CNS, characterized by neuroinflammatory changes (Semmler *et al*, 2005, 2007; Qin *et al*, 2007; Gyoneva *et al*, 2014), identifying that microglia are affected by systemic immune processes. Using *in vivo* 2PLSM, we sought to determine the microglial dynamics behind these changes. Hence, we performed cranial window surgery on 15-month-old (mo) *Cx3cr1*-eGFP$^{-/+}$ mice and 3 weeks later injected them with a single dose of the bacterial cell wall component LPS (1 mg/kg i.p). Following this peripheral challenge, we assessed effects on microglial morphology within the first 48 h post-LPS. 2PLSM revealed that 24 h after LPS injection, microglia cells showed morphological signs of activation, characterized by a significant reduction in the number, length, and maximum order of the branches when compared to control mice (Fig 1A and B). Moreover, we found that changes in microglial morphology peaked at 48 h, with a 50% reduction in all parameters measured (Fig 1A and B). To determine whether these changes were of transient or a more permanent nature and also to assess whether aging, a major priming factor for microglial activation (Cunningham, 2013; Raj *et al*, 2014; Fonken *et al*, 2016), influences these results, 5-month- and 15-month-old *Cx3cr1*-eGFP$^{-/+}$ mice were analyzed longitudinally at 2 and 10 days post-LPS (Fig 1C). Comparison

between 5 and 15mo showed that 15mo animals already presented signs of microglia activation (Fig 1D and E) prior to any LPS administration, which was defined by a significant reduction in the number of processes as well as a reduction in the length of processes and maximum branch order (Fig 1D and E). In addition, we found that as a consequence of the above-described changes in the branches, the brain volume covered (total processes length/cell volume) was significantly reduced at 15mo compared to 5mo already in the absence of any LPS challenge, suggesting that age compromised microglial surveying functions already. As shown for 15mo mice, 5mo animals also had a significant reduction in the morphological parameters analyzed 2 days upon LPS challenge. Interestingly, we found that there was a significant increase for all the morphological parameters 10 days after immune challenge compared to 2 days after LPS injection for 5mo mice (Fig 1D and E). In the case of the 15mo mice, there were no morphological changes between 2 and 10 days post-LPS in the morphological parameters, but also the parameters were indistinguishable from baseline conditions (Fig 1D and E). In order to corroborate the changes observed on the morphological level, microglia activation marker CD68 (Hickman *et al*, 2013) was evaluated by immunohistochemistry. A transient increase in CD68 immunoreactivity was observed 2 days after immune challenge in 15mo mice (Fig EV1). Importantly, 10 days after LPS injection, immunoreactivity levels were the same as PBS-treated mice (Fig EV1).

## NLRP3 ko mice are refractory to peripheral immune challenge and age-associated changes

Inflammasomes form in response to microbial or danger signals, which leads to the cleavage of pro-caspase-1 into the active caspase-1 enzyme. Active caspase-1 then cleaves the pro-forms of the inflammatory cytokines, IL-1β and IL-18, into their active forms (Vanaja *et al*, 2015; Man *et al*, 2016). To assess if the NLRP3 inflammasome is involved in the observed changes during aging and LPS challenge, *Nlrp3*$^{-/-}$ mice were crossed with *Cx3cr1*-eGFP$^{-/+}$ mice to assess microglial dynamics using 2PLSM (Fig 2A). No morphological differences between 5 and 15mo animals were observed at baseline for the assessed parameters, suggesting that *Nlrp3* deficiency protects against age-induced microglial alterations (Fig 2B and C). Moreover, microglia from *Nlrp3*$^{-/-}$ mice were refractory to LPS injection (Fig 2B and C), since no morphological changes were observed in any of the tested time points after LPS challenge. In line with these findings, no changes were observed in the levels of CD68 immunoreactivity after peripheral immune challenge (Fig EV1).

The activation of the inflammasome and subsequent release of IL-1β requires the adaptor protein ASC, which in turn leads to the recruitment of the effector caspase-1 (Baroja-Mazo *et al*, 2014). Interaction of ASC with caspase-1 could lead to the formation of an ASC speck (Venegas *et al*, 2017). We found that 2 days post-LPS challenge leads to an increase in the formation of ASC specks in 15mo mice concordant with inflammasome activation (Appendix Fig S1A). Notably, at 10 days after peripheral immune challenge, a reduction in ASC speck was observed (Appendix Fig S1A and C). It is important to mention that ASC speck formation was dependent on the activation of NLRP3 inflammasome, since no specks were detectable in *Nlrp3*$^{-/-}$ mice (Appendix Fig S1A and C).

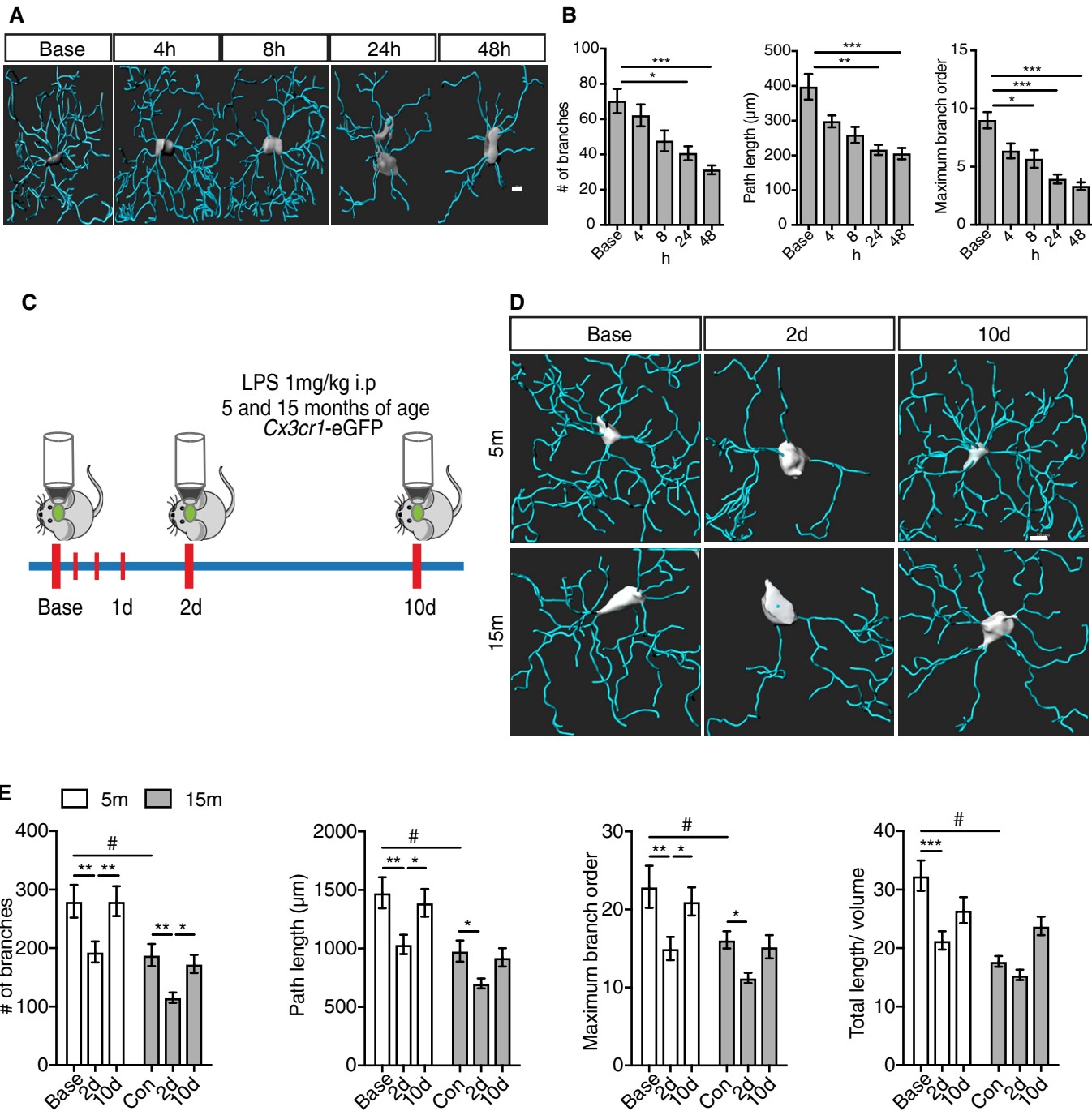

**Figure 1. Systemic inflammation transiently affects microglia in an age-dependent manner.**

A  Representative 3D microglia reconstructions showing morphological changes upon LPS injection within the first 48 h. Scale bar: 10 μm.

B  Morphological parameters quantification within 48 h after LPS injection (mean of 5–6 ± SEM; one-way ANOVA followed by Tukey's *post hoc* test, *$P < 0.05$, **$P < 0.01$, ***$P < 0.001$).

C  Two-photon microscopy experimental design.

D  Representative 3D microglia reconstructions from 5- and 15-month-old mice showing microglia changes 2 and 10 days post-LPS injection. Scale bar: 20 μm.

E  Morphological parameters quantification for 5- and 15-month-old mice after LPS injection (mean of 5–6 ± SEM; two-way ANOVA followed by Tukey's *post hoc* test, ##*$P < 0.05$, **$P < 0.01$, ***$P < 0.001$).

Accordingly, no changes in IL-1β levels were observed in *Nlrp3*$^{-/-}$ mice after LPS injection (Fig EV2A), whereas wild-type mice exhibited an increase 2 days after peripheral challenge, and then, consistent with morphological results, we observed a return to basal levels by 10 days post-LPS challenge. When TNF-α levels were measured, it was found to increase in all the experimental groups,

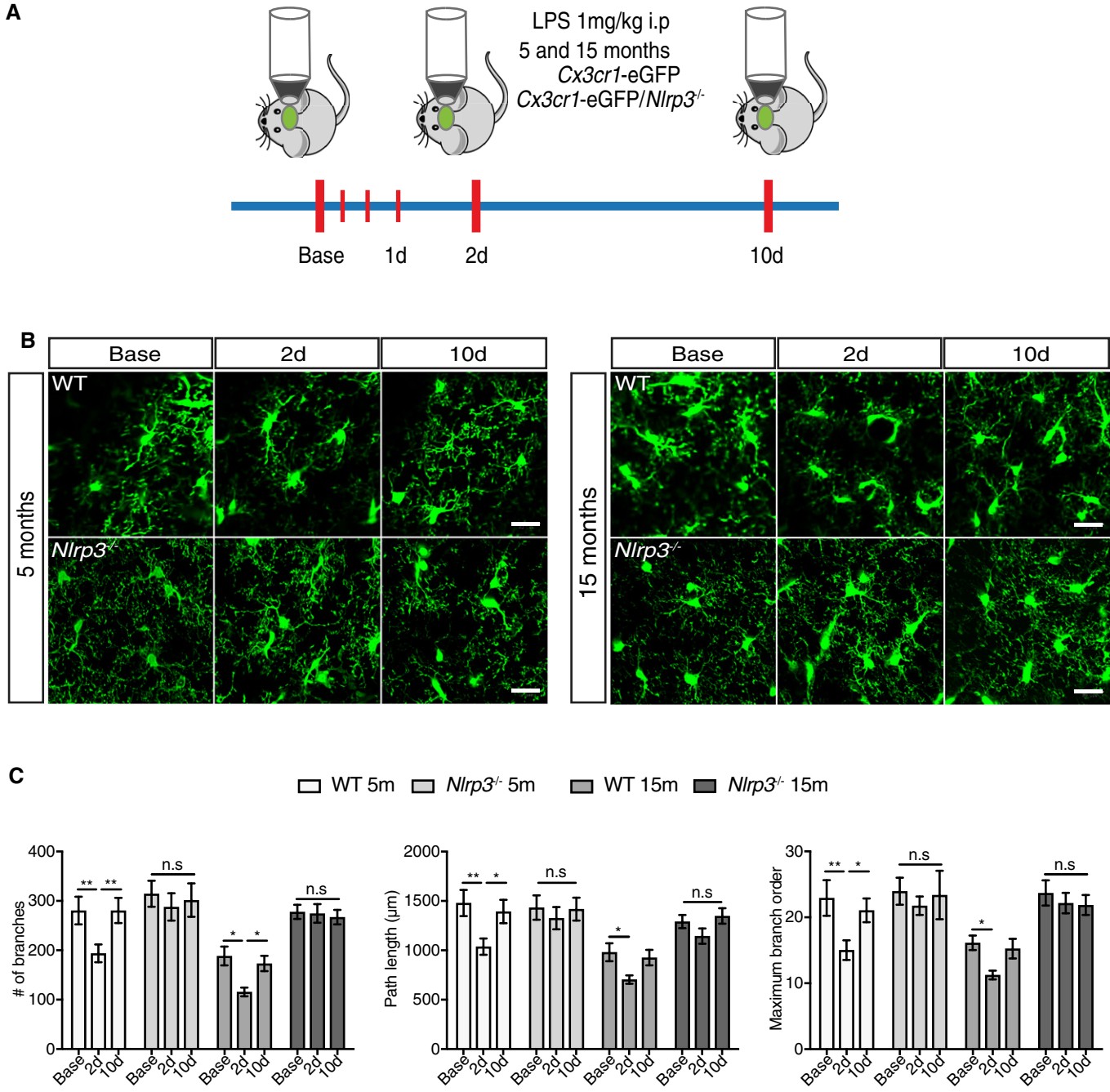

**Figure 2. *Nlpr3* knockout mice are refractory to peripheral immune challenge or age-associated changes.**

A Schematic representation of two-photon microscopy experimental design.

B Two-photon representative images of wild-type and *Nlrp3*$^{-/-}$ mice (5 and 15 months old). Scale bar: 20 μm.

C Quantification of morphological parameters for wild-type and *Nlrp3*$^{-/-}$ mice after LPS injection (mean of 5-6 ± SEM; two-way ANOVA followed by Tukey's *post hoc* test, *$P < 0.05$, **$P < 0.01$).

2 days after peripheral challenge (Fig EV2A). Importantly, this increase was transient; since 10 days after LPS injection, TNF-α levels were found to return to basal levels (Fig EV2A). Remarkably, the liver, as a measurement of the peripheral response, mirrored the immune response observed in the brain but with higher levels of both cytokines (Fig EV2B).

**Astrocytes are transiently activated by LPS peripheral injection**

Astrocytes are CNS cells that participate in a myriad of processes, mainly providing trophic support to neurons and promoting synapse formation and elimination (Jäkel & Dimou, 2017). It has recently been shown that a subset of astrocytes become reactive by

neuroinflammatory microglia (Liddelow *et al*, 2017). Considering the microglial response to peripheral immune challenge, it is plausible to imagine that astrocytes could undergo a process of reactive astrocytosis. Indeed, we found increased GFAP immunoreactivity 2 days after LPS challenge (Fig EV3A and B) in both, 5- and 15-month-old wild-type mice. However, 10 days after peripheral injection, immunoreactivity levels of GFAP were similar to PBS-treated mice (Fig EV3A and B). In line with these results described, no changes in GFAP immunoreactivity were observed in *Nlrp3*$^{-/-}$ mice after LPS injection (Fig EV3A and B), corroborating that neuroinflammatory microglia are required to promote reactive astrocytosis.

### Peripheral immune challenge affects amyloid deposition in APP/PS1 mice

Since systemic inflammation represents a risk for developing neurodegeneration particularly for AD (for review see: Heneka *et al*, 2015), we analyzed the effects of a peripheral immune challenge on pathological hallmarks of AD using APP/PS1 mice. Additionally, we tested whether these effects were mediated by the NLRP3 inflammasome. Therefore, APP/PS1 and APP/PS1/*Nlrp3*$^{-/-}$ mice underwent the same experimental protocols as described above for non-APP/PS1 mice (Fig 1C). While number and size of Aβ deposits were increased in APP/PS1 compared to APP/PS1/*Nlrp3*$^{-/-}$ mice (Fig 3A and B, and Appendix Fig S2A and B) at 15mo, APP/PS1 but not APP/PS1/*Nlrp3*$^{/-}$ revealed a significant increase in Aβ deposition upon LPS challenge at both time points investigated (Fig 3A and B). At 5mo, the time Aβ deposits begin to appear in this model, and no apparent differences were detectable (Maia *et al*, 2013). These results were confirmed by ELISA measurements of Aβ$_{1-40}$ and Aβ$_{1-42}$ (Fig 3B). Of note, this increase was not caused by any modification in the APP processing machinery (Appendix Fig S2C and D). Of note, we observed an increase in the number of ASC specks at both 5 and 15mo of age 2 days post-LPS challenge (Appendix Fig S1B and D). It is also important to mention that as previously described (Venegas *et al*, 2017), ASC specks were also observed in PBS-treated mice (Appendix Fig S1B and D). Moreover, these observations are appeared to be NLRP3-dependent, since no changes in ASC speck formation were observed in APP/PS1/*Nlrp3*$^{/-}$ (Appendix Fig S1B and D). Together, these results suggest that a LPS elicited peripheral immune challenge affects amyloid deposition in aged APP/PS1 mice in an NLRP3-dependent manner.

### Microglia dynamics depend on distance to Aβ deposition in APP/PS1 mice

Microglia cluster around amyloid plaque deposits with their processes being retracted and less dynamic as compared to plaque-free or distant microglia (Condello *et al*, 2015). This suggests that on a functional level, at least two different populations of microglial cells have to be distinguished by location in murine AD models, those cells located near Aβ, and cells more distantly located.

Analysis by 2PLSM revealed that LPS did not lead to further morphological changes in microglia located in the vicinity of Aβ deposits (Fig 4A and B), which is defined as the microglia cells in a 60 μm radius form the amyloid deposit core. This observation is most likely due to the already existing high-level of activation by Aβ itself (Appendix Fig S3). However, the peripheral LPS challenge impaired microglial uptake of Aβ in APP/PS1 but not APP/PS1/*Nlrp3*$^{-/-}$ mice (Fig 4C and D, and Appendix Fig S4). In this sense, the presence or absence of the NLRP3 inflammasome proofed to be a determinant factor, since we have found a significant interaction between time after LPS injection and strain of the mice ($F = 6.44$. DFn = 2 DFd = 13 and a $P$ value of 0.014). Altogether, these results suggest that systemic inflammation affects the functional status particularly the Aβ clearance capacity of these cells.

Since a functional role of beclin-1 for microglial Aβ clearance had been previously demonstrated, we analyzed whether systemic inflammation would affect beclin-1 expression (Lucin *et al*, 2013). In the present study, we found decreased expression of beclin-1 in APP/PS1 but not APP/PS1/Nlrp3$^{-/-}$ mice challenged with LPS (Fig EV4).

In contrast to plaque-associated microglia, the morphological dynamics of plaque-distant microglia were more reminiscent to non-APP/PS1 microglia, with respect to their time-dependent morphological changes after peripheral immune challenge, age, and presence of the NLRP3 inflammasome. In 5mo APP/PS1 mice, the number of branches and the maximum branch order were reduced 2 days post-LPS, followed by a recovery in the number of branches by 10 days (Fig 5A and B). Of note, plaque-distant microglial dynamics in aged APP/PS1 (15mo) mice showed no changes upon peripheral immune challenge. These cells, despite not being in direct contact to Aβ deposits, already exhibited morphological signs of microglial activation (Fig 5A and B), likely to be caused by the presence of soluble Aβ species or inflammatory factors, which may render these cells less responsive to any further immune stimulation. In case of APP/PS1/*Nlrp3*$^{-/-}$ and in line with the results shown for *Nlrp3*$^{-/-}$, we observed that these mice were largely refractory to peripheral immune challenge and aging, since no morphological changes were observed upon LPS challenge.

### Peripheral myeloid cells infiltrate brains of APP/PS1 mice upon peripheral LPS challenge

Myeloid cell infiltration into the brain occurs in several models of inflammation (Wohleb *et al*, 2014; Jay *et al*, 2015; Lévesque *et al*, 2016; Wattananit *et al*, 2016). To investigate this phenomenon, brain sections were immunostained for Iba-1 and the peripheral myeloid marker CD169 (Rice *et al*, 2015; Perez *et al*, 2017; Shinde *et al*, 2018). CD169 has been suggested to be a specific marker for bone marrow-derived monocytes and therefore may help to differentiate microglia from infiltrating myeloid cells (Butovsky *et al*, 2012).

Analysis of young and aged WT and *Nlrp3*$^{-/-}$ mice did not reveal myeloid cell infiltration into the brain upon LPS (Appendix Fig S5A and B). Similarly, LPS did not induce myeloid cell infiltration in young or aged APP/PS1 or APP/PS1/*Nlrp3*$^{-/-}$ mice in brain areas free of Aβ deposits (Appendix Fig S5C and D). Likewise, young APP/PS1 and APP/PS1/*Nlrp3*$^{-/-}$ showed no CD169 immunopositive cells in response to LPS. In strong contrast, CD169-positive cells became detectable in aged, 15mo APP/PS1 mice at 2 days post-LPS and were mostly located in close vicinity to Aβ deposits (Fig 6A and B). Interestingly, this was not found

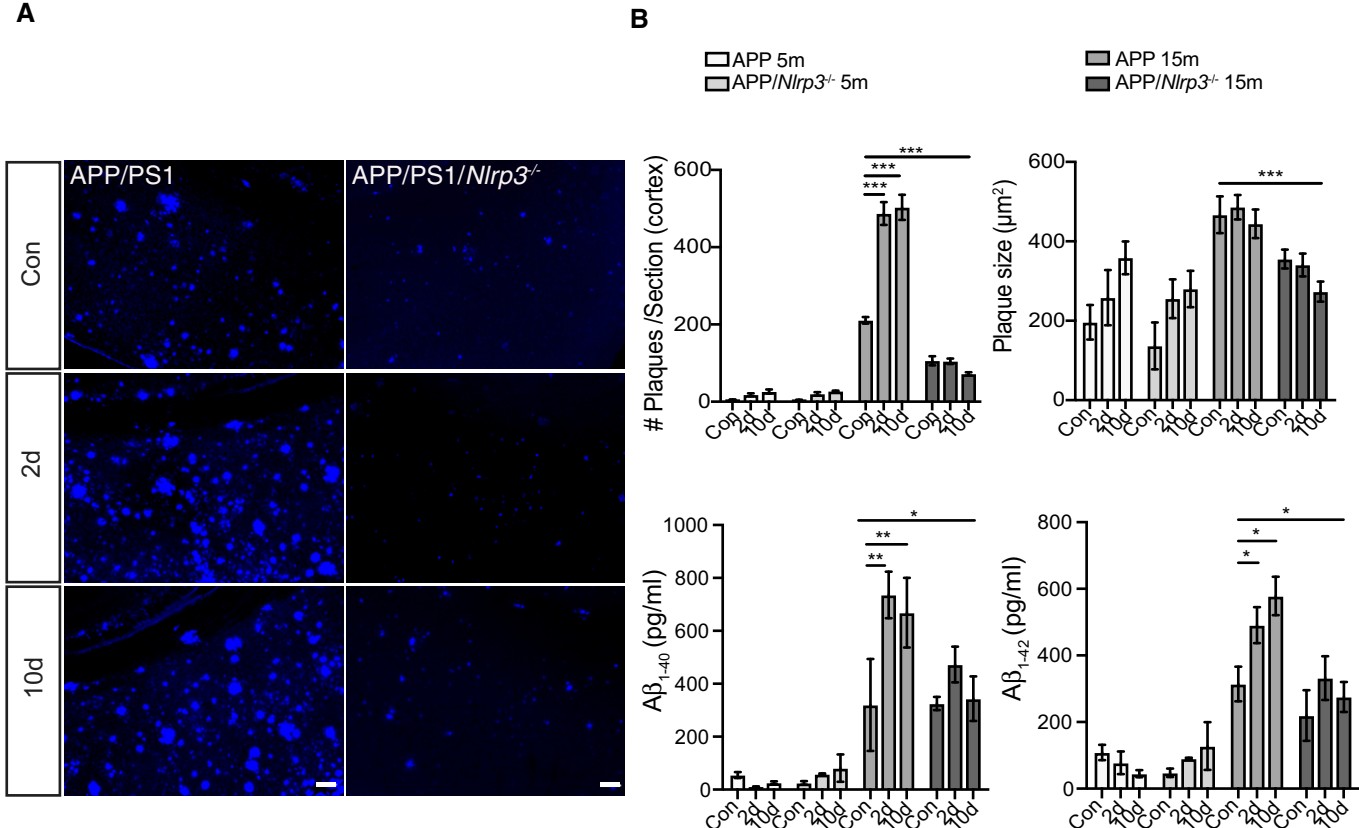

**Figure 3. Peripheral immune challenge affects amyloid deposition in APP/PS1 mice.**

A Representative cortical images of MXO4 staining for APP and APP/$Nlrp3^{-/-}$ 15-month-old mice. Scale bar: 50 μm.

B Cortical amyloid plaque number and size quantification, and Amyloid-beta$_{1-40}$ and $_{1-42}$ ELISA quantification for APP and APP/$Nlrp3^{-/-}$ mice (5- and 15-month-old) (mean of 8 ± SEM; two-way ANOVA followed by Tukey's *post hoc* test, *$P < 0.05$, **$P < 0.01$, ***$P < 0.001$).

in APP/PS1/$Nlrp3^{-/-}$, where no CD169-positive cells were detectable (Fig 6A and B). In order to understand if this infiltration was triggered by the leakage of the blood–brain barrier, fibrinogen was used as a marker (Merlini *et al*, 2019). As a consequence of a compromised blood–brain barrier, fibrinogen is deposited as insoluble fibrin (Davalos *et al*, 2012). Immunohistochemical analysis revealed a significant increase in the amount of fibrinogen present in the brains of APP/PS1 15mo mice upon LPS injection (Fig EV5). It is important to mention that we found no increase whatsoever in fibrinogen in APP/PS1/$Nlrp3^{-/-}$ brains at 15mo (Fig EV5) and not surprising there is a dramatic and significant decrease in fibrinogen when quantified from the brains of APP/PS1/$Nlrp3^{-/-}$ compared to APP/PS1.

Together, this may suggest that immunoattracting mediators, such as cytokines and chemokines, are being produced close to Aβ deposits in aged APP/PS1 mice. Therefore, we hypothesize that $Nlrp3$ knockout may prevent the infiltration of Aβ-directed migration at several levels including the reduction of inflammatory mediators, the capability of peripheral immune cells to enter the brain, and also by protecting the integrity of the blood–brain barrier. Nevertheless, further research is required in order to elucidate the contribution of peripheral cells in this particular context.

## Microglia proliferate upon peripheral immune challenge

The microglia cell population renews physiologically by proliferation in mice and men (Askew *et al*, 2017; Réu *et al*, 2017). In AD and related mouse models, microglial proliferation seems to be accelerated (Olmos-Alonso *et al*, 2016). To understand if a peripheral inflammatory stimulus further increases microglial proliferation, we analyzed cortical and hippocampal sections by immunostaining for Iba-1 and the proliferation marker Ki-67. PBS-treated WT and $Nlrp3^{-/-}$ animals showed only few proliferating microglia (Fig 7A and C), except for 15mo WT mice, that revealed proliferating microglia in the hippocampus (Fig 7B and C). In cortical areas, LPS triggered microglia proliferation in both WT and $Nlrp3^{-/-}$ mice (5 and 15mo), although on a higher proliferation rate in WT compared to $Nlrp3^{-/-}$ animals (Fig 7). In the hippocampus, this difference was only observed when WT and $Nlrp3^{-/-}$ animals were compared at 15mo. Interestingly, hippocampal analysis of proliferative microglia in $Nlrp3^{-/-}$ mice revealed that proliferation was higher at the age of 5 months compared to 15 months old animals (Fig 7B and C).

In contrast to non-APP mice (Fig 7), APP/PS1 and APP/PS1/$Nlrp3^{-/-}$ mice already showed proliferating microglia in the cortex and hippocampus of PBS-treated mice, mainly located at Aβ deposits (Fig 8A and B). Of note, in the absence of LPS, microglia

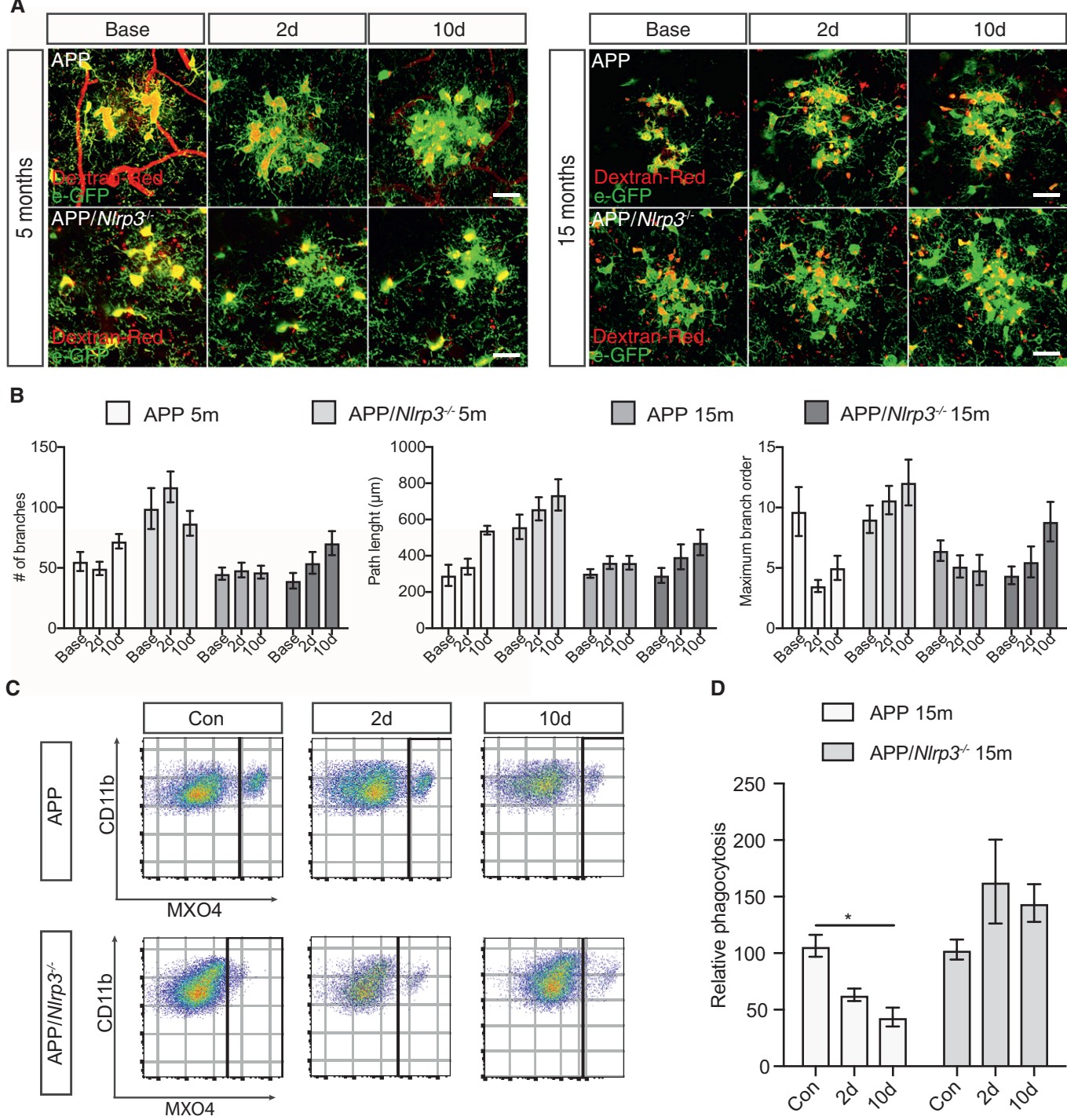

**Figure 4. Microglia dynamics depend on distance to Aβ deposition in APP/PS1 mice.**

A Two-photon images of microglia (eGFP) cells clustering around amyloid plaque for APP and APP/Nlrp3$^{-/-}$ (5 and 15 months old). Scale bar: 20 μm.

B Quantification of morphological parameters in (A) (mean of 5–6 ± SEM).

C Flow cytometry plots from APP and APP/Nlrp3$^{-/-}$ mice (15 months old), cells were gated on CD11b and MXO4 after microglia isolation.

D Relative Aβ microglia uptake quantification (mean of 5 ± SEM; two-way ANOVA followed by Tukey's *post hoc* test, *P < 0.05).

proliferation was higher in APP/PS1 brains compared to APP/PS1/ Nlrp3$^{-/-}$. Importantly, peripheral immune challenge triggered microglia proliferation regardless of the genotype, 2 days post-

challenge (Fig 8A–C), except in cortical areas of APP/PS1 at 5 months of age. For this particular group, LPS injection did not lead to an increase in microglia proliferation (Fig 8A and C).

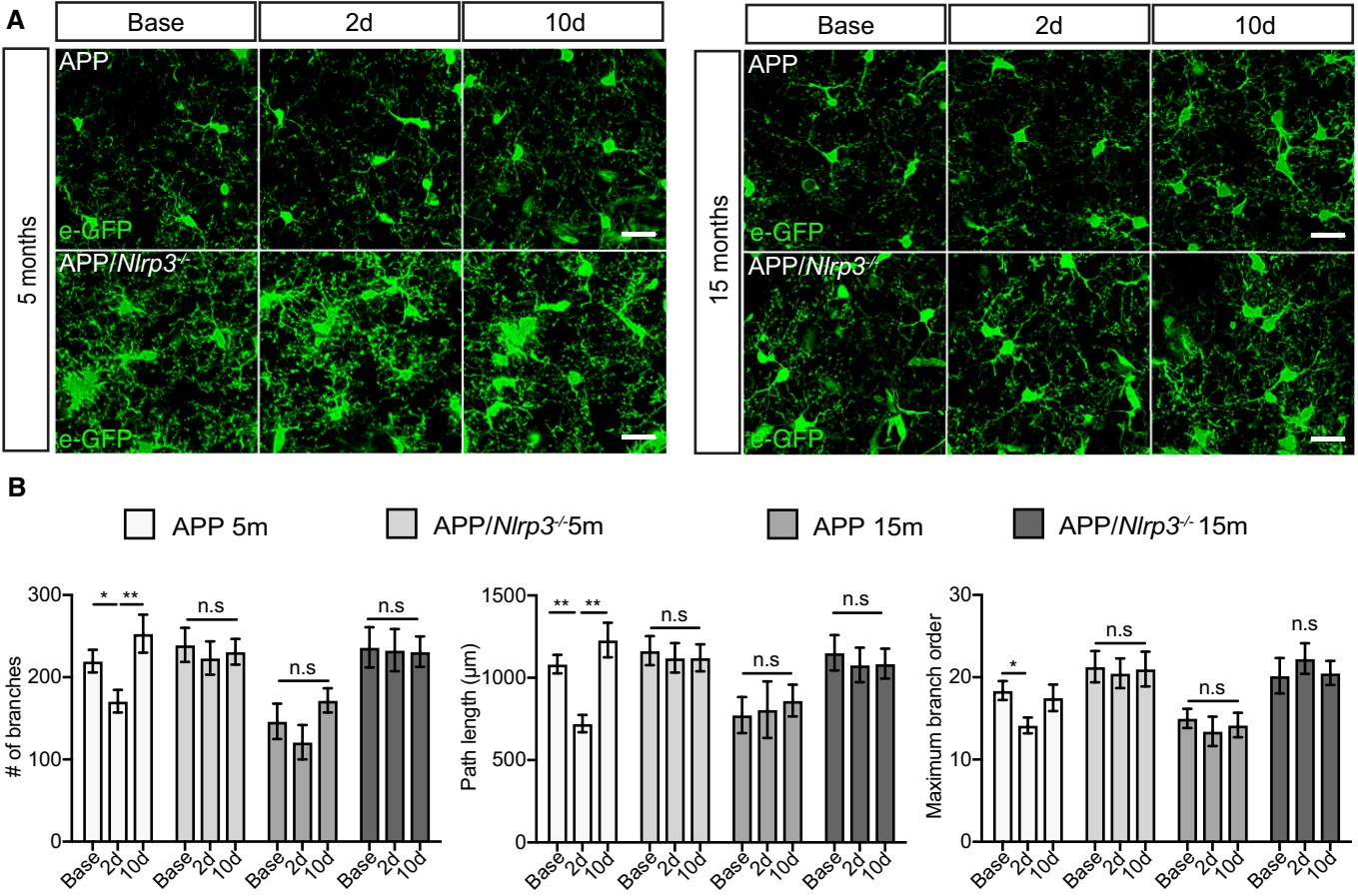

**Figure 5. Microglia dynamics depend on distance to Aβ deposition in APP/PS1 mice.**

A   Two-photon microglia (eGFP) images from APP and APP/*Nlrp3*⁻/⁻ in areas free of plaque (5 and 15 months old mice). Scale bar: 20 μm.

B   Quantification of APP and APP/*Nlrp3*⁻/⁻ morphological parameters for microglia in areas free of plaque (mean of 5 ± SEM; two-way ANOVA followed by Tukey's *post hoc* test, \*P < 0.05, \*\*P < 0.01).

Considering that proliferative microglia were mostly found in areas of amyloid deposition, a possible explanation for this observation could be that low number of amyloid deposits found in APP/PS1 5mo mice could cause this variation.

## Discussion

As the sentinels of the brain, microglia are extremely versatile to external stimuli. Physiologically, they exhibit a high order of ramification, but at the same time remain incredibly motile in order to continually survey and maintain homeostasis in their environment. In the context of aging and neurodegeneration, however, their morphology and function drastically changes (Ransohoff & Perry, 2009). Indeed, branches are retracted and their order reduced, while at the same time the soma volume increases (Kreutzberg, 1995). Functionally, these cells acquire a pro-inflammatory phenotype, releasing inflammatory cytokines and other neurotoxic mediators. In the present study, we assessed microglial dynamics in the context of systemic inflammation and Aβ deposition by 2PLSM in wild-type and APP/PS1 transgenic mice, a murine model of AD. A

3-dimensional analysis and quantification of microglia revealed that systemic inflammation: (i) transiently affects microglia in an age-dependent manner, (ii) worsens Aβ deposition by affecting microglial clearance, and (iii) increases microglial proliferation as sign of disease acceleration. Most importantly, we have identified the activation of the NLRP3 inflammasome as a key mediator of these effects.

In this work, we show microglial dynamics *in vivo* for the first time upon systemic inflammation in the context of aging and neurodegeneration. The two-photon system represents a major advantage allowing the possibility of longitudinally analyzing microglia cells in their environment within an intact brain. Additionally, we should also emphasize that this method significantly reduces the number of mice, considering that the same mice were analyzed longitudinally during all imaging sessions.

Microglial cell dynamics were strongly impaired at 48 h after peripheral immune challenge in wild-type animals. This phenomenon was characterized by a 50% reduction in the number of branches, bifurcations, path length, and maximum branch order. Together, these changes indicate a substantial loss of surveillance capacity in line with similar findings by previous

## Cortex

## Hippocampus

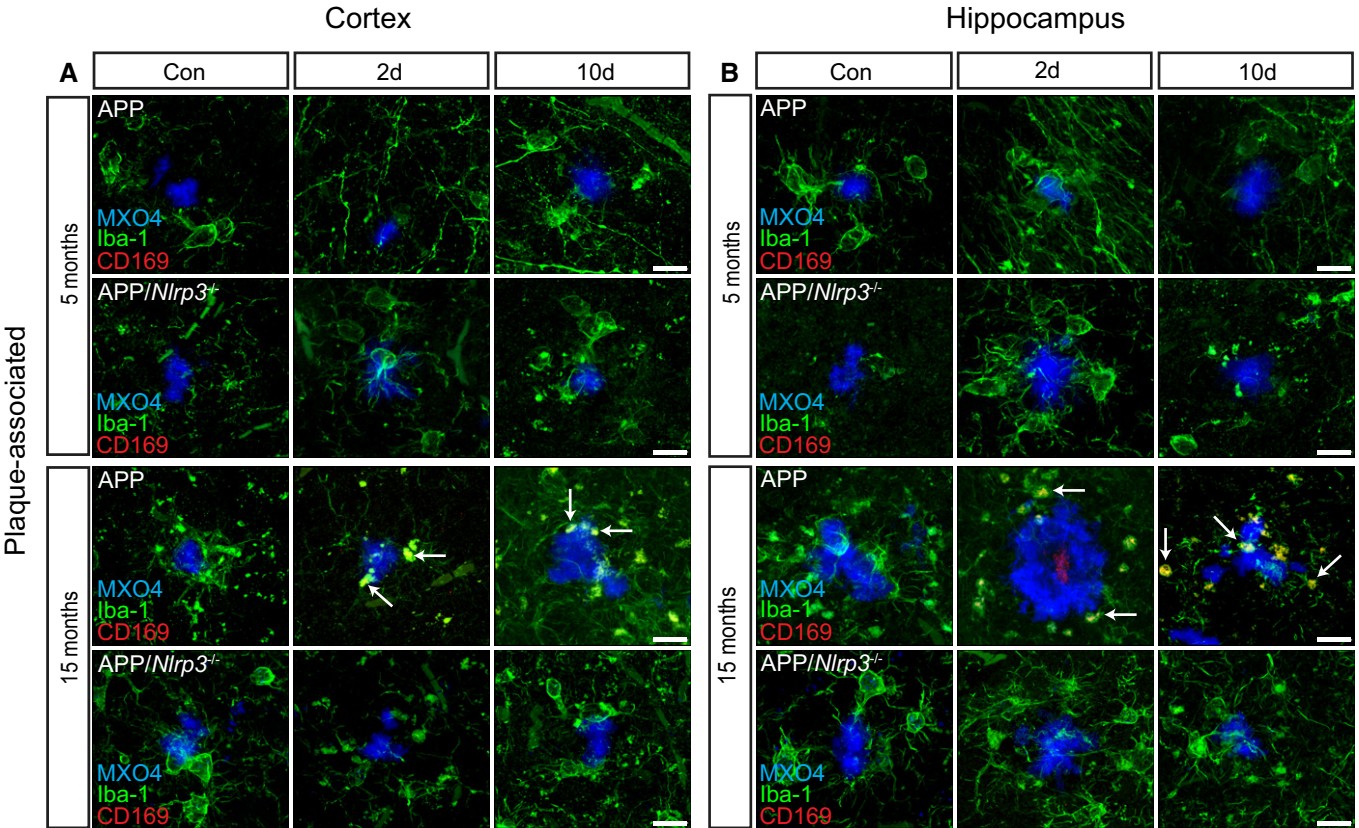

**Figure 6. Peripheral myeloid cells infiltrate into APP/PS1 mice upon LPS injection.**

A, B    Iba-1 (green), CD169 (red), and MXO4 staining in plaque-associated areas (cortex and hippocampus) of 5 and 15 months old of APP and APP/*Nlrp3*$^{-/-}$. Note the colocalization between Iba-1 and CD169 (white arrows) in APP 15-month-old mice 2 days post-LPS injection. Scale bar: 20 μm.

immunohistochemical studies (Qin *et al*, 2007; Gyoneva *et al*, 2014; Jo *et al*, 2017). Since microglia may undergo age-dependent changes affecting their reaction to external stimuli, mice were studied at 5 (adult) and 15 months (aged) of age. Aging alone caused morphological changes in microglia, characterized by a reduction of all parameters assessed, supporting the hypothesis that age-associated brain changes may act as a priming factor for microglia. In both adult and aged wild-type animals, changes in microglial dynamics induced by systemic inflammation were of transient nature, peaking at 48 h and returning back to basal levels at 10 days after LPS challenge. In contrast to young animals, aged mice started from a primed, pre-activated level, but also returned completely back to baseline at 10 days, suggesting that aging affects the basal activation state, but not the capacity to fully recover, at least not at the time points and ages assessed in this study. These findings from dynamic morphological *in vivo* imaging parallel data from transcriptional analysis which revealed age-related changes in genes encoding for immune regulation, cytokine secretion, immune adhesion, and chemotaxis (Grabert *et al*, 2016). In addition, genes known as "off-signals" (Biber *et al*, 2007), including *Cd200* and *Cd300,* were downregulated in aged microglia. These results suggest that changes in microglial morphology are accompanied by functional responses, such as cytokine release as well as impaired cerebral metabolism (Semmler *et al*, 2008) or iNOS expression (Weberpals *et al*, 2009).

Since systemic inflammation has been identified as a risk factor for persisting cognitive deficits in humans (Iwashyna *et al*, 2010; Semmler *et al*, 2013; Walker *et al*, 2017) and likewise for the development of neurodegenerative diseases such as AD (Widmann & Heneka, 2014), we tested whether a single peripheral immune stimulus would affect neuropathological changes such as Aβ pathology and neuroinflammation in a murine AD model.

A single LPS injection was sufficient to cause an increase in cerebral Aβ deposition in aged APP/PS1 transgenic mice. Since no changes in APP processing were detectable, we hypothesized that microglia Aβ clearance might be affected by peripheral inflammation. In line with this assumption, LPS was found to reduce microglial Aβ uptake. In parallel, beclin-1, a factor involved in microglial Aβ phagocytosis (Lucin *et al*, 2013), was reduced after peripheral immune challenge, supporting the hypothesis that dysfunctional microglia may account for the increase of Aβ deposition. These results are in concordance with previous reports showing that peripheral administration of LPS can increase amyloid deposition in transgenic murine models of amyloidosis (Lee *et al*, 2008; Joshi *et al*, 2014; Wendeln *et al*, 2018).

Recent evidence suggests that the microglia population is not homogeneous within the AD brain (Baron *et al*, 2014; Gyoneva *et al*, 2016; Keren-Shaul *et al*, 2017). In concordance with these previous findings, we have identified two different populations

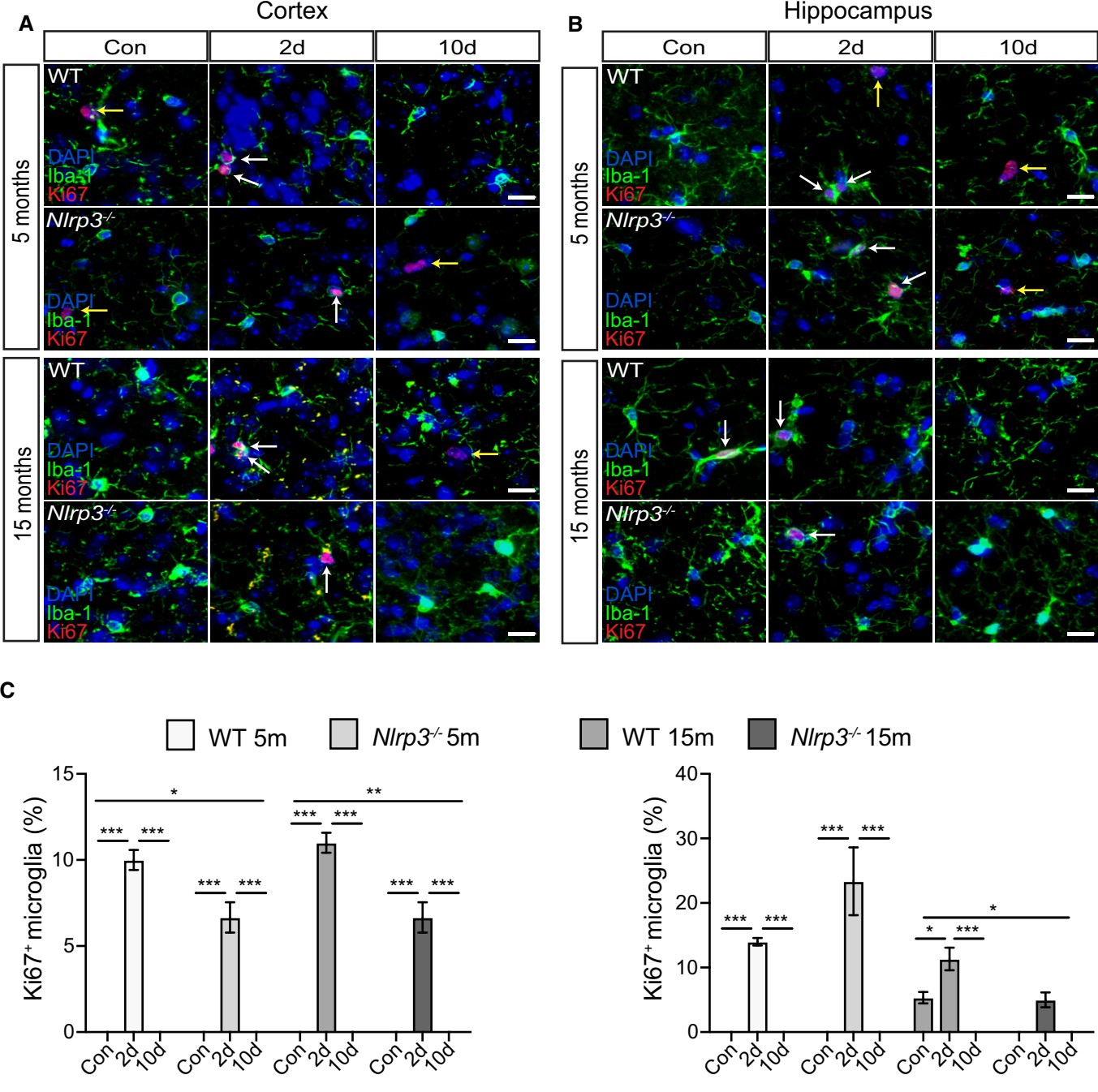

**Figure 7. Microglia proliferate in non-APP mice upon peripheral immune challenge.**

A, B   Iba-1, Ki67, and DAPI staining in the cortex and hippocampus of wild-type and *Nlrp3*⁻/⁻ (5 and 15 months old). Microglia proliferate upon LPS injection (white arrows). Non-microglia cells proliferation was observed as well (yellow arrows). Scale bar: 20 μm.

C   Quantification of microglial proliferation in cortex (left panel) and hippocampus (right panel) (mean of 5 ± SEM; two-way ANOVA followed by Tukey's *post hoc* test, *P < 0.05, **P < 0.01, ***P < 0.001).

with regard to their distance to Aβ deposits. Our *in vivo* 2PLSM analysis showed that microglia located at the site of Aβ deposition does not show striking morphological changes upon peripheral immune stimulation as compared to microglia located at a greater distance from the plaques. Since microglia at the site of Aβ deposition already show substantial changes with reduction

of branch number, length, and order, we suggest that these cells may not be "further" reacting, at least with respect to morphological measures. Instead, distant microglia showed dynamic changes in morphology, which were similar to non-APP/PS1 mice.

In order to test the impact of innate immune signaling on the observed changes, we analyzed animals carrying a genetic deletion

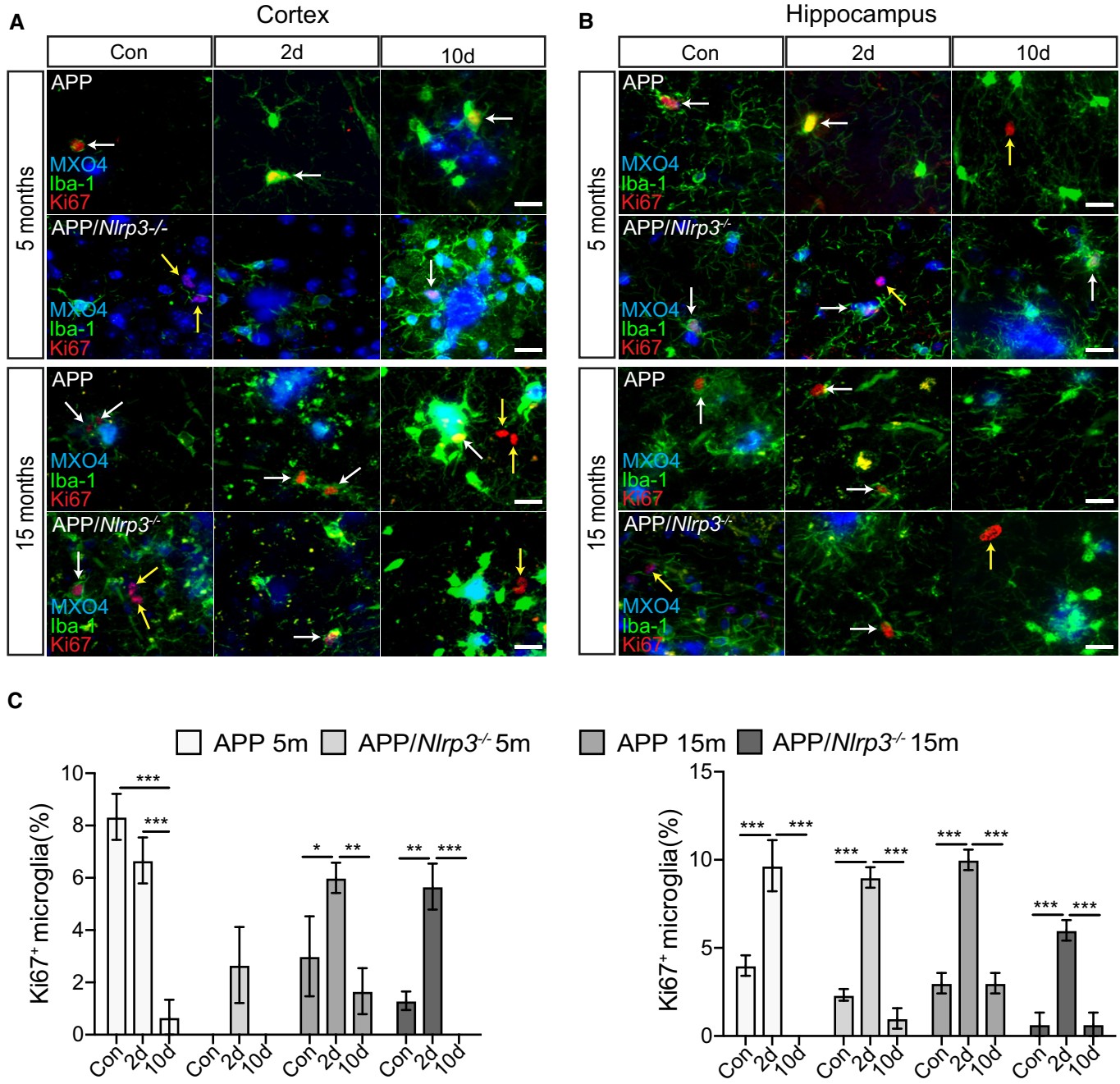

**Figure 8. Microglia proliferate in APP/PS1 mice upon peripheral immune challenge.**

A, B   Iba-1, Ki67, and MXO4 staining in the cortex and hippocampus of APP and APP/*Nlrp3*$^{-/-}$ (5 and 15 months old). Microglia proliferate upon LPS injection (white arrows). Non-microglia cells proliferation was observed as well (yellow arrows). Scale bar: 20 μm.
C   Quantification of microglial proliferation in cortex (left panel) and hippocampus (right panel) (mean of 5 ± SEM; two-way ANOVA followed by Tukey's *post hoc* test, *$P < 0.05$, **$P < 0.01$, ***$P < 0.001$).

of the *Nlrp3* gene, thus blocking NLRP3 inflammasome activation, a central pathway of peripheral and cerebral innate immunity. In general, inflammasomes are subcellular multiprotein complexes that play an important role in host defense against extracellular, vacuolar, and intracellular bacteria, fungi, and viruses (Cassel *et al*, 2009; Lamkanfi & Dixit, 2014; Song *et al*, 2017). Given the fact that the

NLRP3 inflammasome has been implicated in peripheral immune reactions upon bacterial challenge and also has been shown to contribute to neuroinflammatory changes (Heneka *et al*, 2013; Lee *et al*, 2013; Grace *et al*, 2016; Yin *et al*, 2017) made it a likely candidate for LPS-induced changes during aging and Aβ deposition. Microglia from *Nlrp3*-deficient animals were refractory to aging- and

inflammation-induced morphological changes, at least with respect to the current analysis. In line with the morphological analysis, no changes in the activation marker CD68 immunoreactivity were observed upon LPS injection. Nevertheless, we cannot exclude that other markers could be affected.

This is consistent with a previous study showing that ablation of the NLRP3 inflammasome controls age-related inflammation (Youm *et al*, 2013).

Furthermore, our results indicate that the NLRP3 inflammasome is involved in the changes of Aβ deposition upon peripheral immune challenge. In line with previous findings, APP/PS1/*Nlrp3*$^{-/-}$ mice showed significantly less Aβ deposition compared to APP/PS1 animals (Heneka *et al*, 2013). Of note, peripheral immune challenge did not aggravate Aβ deposition in *Nlrp3*-deficient mice. While this study was not designed to analyze whether peripheral or central *Nlrp3* deficiency is mediating this beneficial effect, one contributing mechanism could be a preservation of Aβ clearance mechanisms as previously reported (Heneka *et al*, 2013). This is supported by the observation that no changes in phagocytic capacity or beclin-1 were observed in APP/PS1/*Nlrp3*$^{-/-}$ after LPS injection unlike APP/PS1 mice. Thus, NLRP3 inhibition may represent a novel therapeutic candidate for brain protection during systemic inflammation.

# Materials and Methods

### Reagents

Ultrapure LPS (*S. typhimurium*) was obtained from Sigma-Aldrich (Darmstadt, Germany, Cat. No. L6143). Antibodies to Iba-1 were purchased from Wako Chemicals (Neuss, Germany, Cat. No. 019-19741) (pAb rabbit anti-mouse 1:500) and Novus Biologicals (Wiesbaden, Germany, Cat. No. NB100-1028) (pAb goat anti-mouse 1:50). Ki67 antibody was obtained from Abcam (Cambridge, UK, Cat. No. ab15580) (pAb rabbit anti-mouse 1:500). CD169 antibody was purchased from Bio-Rad (Munich, Germany, Cat. No. MCA884) (mAB rat anti-mouse 1:50). Fibrinogen antibody was purchased from Abcam (Cambridge, UK, Cat. No. ab34269) (pAb rabbit anti-mouse 1:200). CD68 and GFAP antibodies were obtained from Thermo Fisher (Waltham, MA, US, Cat. No. NC9471873 and 13-0300, respectively) (mAb rat anti-mouse 1:500 and mAb rat anti-mouse 1:1,000, respectively). ASC antibody was purchased from AdipoGen (Liestal, Switzerland, Cat. No. AL177, pAb rabbit anti-mouse 1:300).

### Study approval

Animal care and handling was performed according to the declaration of Helsinki and approved by the local ethical committees (animal experimentation project number: 84.02.04.2013.A101).

### Animals

C-X3-C motif, receptor 1-GFP (*Cx3cr1*-eGFP) transgenic mice (Jung *et al*, 2000) exhibiting a microglia-specific expression in CNS and APP/PS1 transgenic animals (Jankowsky *et al*, 2001) were purchased from The Jackson Laboratory (Bar Harbor, ME) on a C57BL/6 background. *Nlrp3*-deficient animals (Kanneganti *et al*,

2006) were also backcrossed onto C57BL/6. All mice were housed under standard conditions at 22°C with a 12-h light/dark cycle and free access to food and water. The following animal groups were analyzed: *Cx3cr1*-eGFP/$^{+}$, *Cx3cr1*-eGFP/$^{+}$/*Nlrp3*$^{-/-}$, *Cx3cr1*-eGFP/$^{+}$/APP/PS1, and *Cx3cr1*-eGFP/$^{+}$/APP/PS1/*Nlrp3*$^{-/-}$ at 5 and 15 months of age.

### Animal treatment and surgery

Animals were intraperitoneally (i.p.) injected with a single dose of LPS (1 mg/kg body weight). For cranial window surgery, animals were anesthetized with 1.5 mg/kg ketamine (Ratiopharm, Ulm, Germany) and 0.1 mg/kg xylazine (Serumwerk Bernburg, Bernburg, Germany). Mice were additionally injected subcutaneously with a mixture of 0.1 mg/kg Buprenorphine hydrochloride (Indivior, Slough, UK), 6 mg/kg dexamethasone (Jenapharm, Jena, Germany), and 5 mg/kg carprofen (Pfizer, Berlin, Germany) to reduce pain and inflammation. Mice were placed on a heating blanket connected to a rectal probe for maintaining body temperature at 37°C. The head was fixed in a stereotaxic frame (Narishige, London, UK), and craniotomy was performed as previously described (Holtmaat *et al*, 2009) with some minor modifications. Briefly, the hair was removed using a hair removal cream; a midline incision was made and the periosteum was gently removed by scraping. A 3 mm window over the somatosensory (−3;0 mm antero-posterior and 0;+3 mm lateral according to Bregma) cortex area was generated using a dental driller (Schick, Schemmerhofen, Germany) with a 0.4 mm drilling head.

A 5 mm coverslip was attached to the skull using cyanoacrylate glue (UHU, Brühl Germany). A customized titanium ring was attached to the skull using dental cement (Heraeus Kulzer, Hanau, Germany). The mice were imaged after a 2-week recovery period.

### *In vivo* two-photon laser scanning microscopy

A Nikon A1R MP microscope and a titanium–sapphire laser (Chameleon Ultra, Coherent, Santa Clara, CA) were used for mouse brain *in vivo* imaging. The same mice were imaged along three different time points (Appendix Fig S5), before (base), 2 and 10 days after LPS injection. The brain vasculature was used as a reference point to longitudinally image the same brain areas. Importantly, none of the mice died during the imaging session.

During all imaging sessions, the laser power did not exceed 30 mW. Three-dimensional *Z*-stacks (40 μm length; 0.5 μm step between optical planes) were acquired using a Nikon 25× objective (1.1 NA). For visualization of blood vessels, 20 mg/kg body weight dextran red 70KDa (Sigma-Aldrich, Darmstadt, Germany) was i.p. injected 30 min before the imaging session. To visualize Aβplaques, 10 mg/kg methoxy-XO4 (Tocris Bioscience, Bristol, UK) in 50% DMSO/50% NaCl (0.9%), pH 12, was i.p. injected 3 h before the imaging session (Bolmont *et al*, 2008).

For quantitative analysis, two-photon *Z*-stacks were automatically reconstructed using a self-customized python-based script (Ativie *et al*, 2018). Reconstructions were visually checked using the ImageJ plugin "simple neurite tracer" (Appendix Fig S6). Each cell was individually extracted, and all the individual files were analyzed using the open source software L-measure (Scorcioni *et al*, 2008).

## Histology and immunohistochemistry

Mice anesthetized with 100 mg/kg body weight ketamine and 16 mg/kg body weight xylazine were transcardially perfused with cold PBS (30 ml), and the brain, lung, liver, spleen, and kidney were removed. One hemisphere of the brain and the remaining organs were flash-frozen in liquid nitrogen and stored at $-80°C$. The other hemisphere was fixed with 4% paraformaldehyde by immersion for 24 h at 4°C, washed three times with cold PBS, and stored in PBS-NaN$_3$. 40 μm coronal, vibratome sections were stained free-floating. For that, sections were washed three times for 5 min with PBS, Triton X-100 0.1% (PBS-T), blocked for 1 h with BSA 1%, and incubated overnight with the primary antibodies. Next day the sections were washed three times for 5 min in PBS-T, incubated with antibody conjugates (1:500) (Invitrogen, Darmstadt, Germany) for 60 min, and washed three times with PBS for 5 min. For visualization of amyloid Aβ deposits, sections from APP mice were incubated with 10 μM methoxy-XO4 for 10 min and then washed three times in PBS. Sections were mounted using Immu-Mount (Thermo Scientific, Bonn, Germany).

## Epifluorescence microscopy

Images were acquired using an Olympus BX61 epifluorescence microscope equipped with a disk-spinning unit. Aβ deposits were imaged using a 4× (0.16 NA) objective. For Ki-67 and CD169, Z-stacks were taken with an oil immersion 40× (1.0 NA) and 60× (1.35 NA), respectively. For quantitative image analysis of hippocampal and cortical immunostaining, serial coronal sections of five animals from each group were examined. For each animal, 10 parallel sections, with a defined distance of 40 μm and showing both the hippocampus and the cortex, were analyzed. Quantification of the number and size of Aβ deposits in the cortex and hippocampus was determined using ImageJ plugin 3D object counter (Bolte & Cordelieres, 2006). Quantification of microglia proliferation was conducted by manual cell counting using merged images of Iba1 and Ki67 staining. Proliferative microglia were normalized to the total number of microglia cells counted. For fibrinogen, CD68 and GFAP and ASC, Z-stacks were taken using a 40× (1.0 NA) glycerin immersion objective on 5 parallel sections per animal. Integrated staining intensity was analyzed using ImageJ. ASC specks were quantified using ImageJ, and the ratio of ASC/Iba-1$^+$ cells was calculated.

## Protein extraction

Brains of 5- and 15-month-old mice were homogenized in PBS containing 1 mM EDTA and 1 mM EGTA and protease inhibitor cocktail, further extracted in RIPA buffer [25 mM Tris–HCl (pH 7.5), 150 mM NaCl, 1% Nonidet P-40, 0.5% NaDOC, 0.1% SDS], and centrifuged at $20,000 \times g$ for 30 min, and the pellet was solubilized in 2% SDS, 25 mM Tris–HCl (pH 7.5). Samples were separated by NuPage and immunoblotted using antibodies, anti-CT20 (1:200, Millipore, Darmstadt, Germany, Cat. No. 171610) and anti-beclin 1 (1:1,000, Cell Signaling, Frankfurt, Germany, Cat. No. 3738) followed by incubation with appropriate secondary antibodies. Immunoreactivity was detected using an Odyssey CLx imager

(Licor, Bad Homburg, Germany), and pictures were analyzed using Image Studio (Licor, Bad Homburg, Germany).

## ELISA quantification of cerebral Aβ concentration

Quantitative determination of amyloid-β was performed using an electrochemiluminescence ELISA for Aβ$_{38}$, Aβ$_{40}$, and Aβ$_{42}$ according to the protocol of the supplier (Meso Scale Discovery, Rockville, MD, USA). Signals were measured on a SECTOR Imager 2400 reader (Meso Scale Discovery, Rockville, MD, USA).

## ELISA pro-inflammatory response quantification

Pro-inflammatory response was determined in brain lysates using the V-PLEX Plus Pro-inflammatory Panel 1 (mouse) Kit for 10 cytokines (IFN-γ, IL-1β, IL-2, IL-4, IL-5, IL-6, KC/GRO, IL-10, IL-12p70, and TNF-α) following the protocol provided by the supplier (Meso Scale Discovery, Rockville, MD, USA). Briefly, 50 μl of diluted sample, calibrator, or control was added per well. The plate was sealed with an adhesive plate seal and incubated at room temperature with shaking for 2 h. Later, the plate was washed three times and the detection antibody was added. The plate was sealed and incubated at room temperature with shaking for 2 h. Finally, the plate was washed and the read buffer was added. Signals were measured on a SECTOR Imager 2400 reader (Meso Scale Discovery, Rockville, MD, USA).

## Isolation of microglia from adult mouse brains

Neural Tissue Dissociation Kit (P), Myelin Removal Beads II, and CD11b (microglia) MicroBeads (Miltenyi Biotec, Bergisch Gladbach, Germany) were used for magnetic isolation of microglial cells from adult APPPS1/*Cx3cr1-e*GFP$^{-/+}$, APPPS1, APPPS1/*Nlrp3*$^{-/-}$/*Cx3cr1-e*GFP/$^+$, and APPPS1/*Nlrp3*$^{-/-}$ mice brains, in accordance with the manufacturer's guidelines. Briefly, brains were dissected after perfusion with PBS, enzymatically digested using the Neural Tissue Dissociation Kit. Cells were incubated for 15 min at 4°C with Myelin Removal Beads II (Miltenyi Biotec, Bergisch Gladbach, Germany) and separated from myelin in a magnetic field using LS columns, MACS MultiStand, and QuadroMACS (Miltenyi Biotec, Bergisch Gladbach, Germany). Cells were incubated with CD11b (microglia) MicroBeads (Miltenyi Biotec, Bergisch Gladbach, Germany) for 15 min followed by separation of CD11b-positive cells in a magnetic field using MS columns and OctoMACS (Miltenyi Biotec, Bergisch Gladbach, Germany).

## Flow cytometry

To detect cell death, microglia were incubated with LIVE/DEAD Fixable Far Red Dead Cell Stain Kit (Thermo Fisher Scientific, Waltham, MA, US). Cells were labeled with PE rat anti-mouse CD11b and APC rat anti-mouse CD45 (BD Bioscience, Drive Franklin Lakes, NJ, USA) diluted 1:50 for 30 min at RT. FMO (Fluorescence Minus One) controls and the isotype control for the CD11b-PE antibody were performed. For our gating strategy (Appendix Fig S7), we first removed doublets cells using the FSC parameters; next, dead cells were removed using a live/dead dye. For our methoxy-XO4 analysis, we took cells that were positive for

both CD11b and CD45. In APP/PS1 mice, it has been described that microglia particularly close to the plaque upregulate CD45 (Maier *et al*, 2008; Keren-Shaul *et al*, 2017). Cells expressing CD45 and not CD11b were excluded, as these could be T and B cells. The specificity of the CD11b antibody signal and the gating strategy was subsequently controlled using APPPS1/*Cx3cr1-e*GFP$^{-/+}$ mice. Samples were measured using a BD FACSCanto™ II (BD Biosciences, Franklin, NJ, USA) and analyzed using FlowJo V10.2 (Ashland, OR, USA). For analysis, the CD11b/CD45 population was gated as previously described (Kummer *et al*, 2012; Heneka *et al*, 2013). WT mice injected with methoxy-XO4 were used to determine the methoxy-X04 threshold for non-phagocytizing cells, and unstained WT cells were used to determine background fluorescence. Finally, methoxy-O4-positive and methoxy-O4-negative populations within CD11b/CD45 were discriminated and the ratio calculated.

## Statistical analysis

Data were analyzed using GraphPad Prism version 6 (GraphPad Software, La Jolla, CA) and presented by mean ± SEM. Differences among the groups were examined using one- or two-way ANOVA followed by Tukey's *post hoc* test. Results were considered to be statistically significant if $P < 0.05$.

Expanded View for this article is available online.

## Acknowledgement

This project was founded by ERA-NET TracInflam (01EW1508) to MTH.

## Author contributions

DT, DM, JMS-C, MH, DG, HS, EL, DG, and MTH performed experiments and analyzed data. DT, HS, EL, DG, and MTH designed the experiments and wrote the paper. Results were discussed, and all the authors commented the manuscript.

## Conflict of interest

The authors declare that they have no conflict of interest.

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
