## [Review Process File · The EMBO Journal]

Systemic inflammation impairs microglial Ab clearance through NLRP3 inflammasome

Dario Tejera, Dilek Mercan, Juan M. Sanchez-Caro, Mor Hanan, David Greenberg, Hermona Soreq, Eicke Latz, Douglas Golenbock, Michael T. Heneka

Review timeline:

Submission date:	8th Nov 2018
Editorial Decision:	21st Dec 2018
Revision received:	7th May 2019
Editorial Decision:	31st May 2019
Revision received:	30th Jun 2019
Accepted:	3rd Jul 2019

Editor: Karin Dumstrei

Transaction Report:

1st Editorial Decision

21st Dec 2018

Thank you for submitting your manuscript to The EMBO Journal. Your study has now been seen by three referees and their comments are provided below.

The manuscript received a bit of a mixed response. The referees appreciate the interest of the topic, but also raise issues regarding novelty and find the manuscript fairly descriptive. I appreciate the issues raised by the referees, but also find the analysis interesting and insightful. Should you be able to address the concerns raised and provide more data to support the key conclusions as well as adding some mechanistic insight into the role of NLRP3 in microglia and how it is activated by LPS (ref #2 and 3) then I would be interested in considering a revised version.

REFeree REPORTS:

Referee #1:

Briefly, LPS given peripherally changed microglial morphology as measured by 2 photon microscopy and made them less able to clear amyloid beta. That change is reported to be NLRP3-dependent.

KEY QUESTION: NLRP3^{-/-} mice are already protected against the APP/PS1 phenotype so we need to be sure that any beneficial effects of the KO are clearly shown to be selective to the LPS-induced changes and not simply explained by the different underlying condition at the time of the LPS challenge.

There are multiple interesting and potentially important observations here, but the key assertion that the effects of LPS on microglial activation, phagocytosis, proliferation and monocyte infiltration are not fully supported by the data as they currently stand. The data need more explicit statistical analysis (throughout) in order to increase confidence in some of the findings and in some cases the

data simply do not support the conclusions. It is very important that the known affect of NLRP3^{-/-} on some of these parameters in APP/PS1 is not mistaken for a NLRP3-dependent effect on LPS on those same parameters.

Results

Figure 1

1f - 'recovery was restricted to normalization of the branch number'. It is not clear what this means, since all parameters seem to recover at 10d back to control levels. It is not clear what statistical justification is used for this, but it is absolutely clear that they are not significantly different from their baseline level. The benefits of real-time and repeated measures allows the authors to see that there are changes at 2 days but by 10 days they are once again indistinguishable from controls: LPS-induced reductions in branch number, path length, and maximum branch order all recover to levels that are indiscernibly different from control levels. To say that these microglia are not different to 2d is entirely counterintuitive. To look at deflections from control levels both post-LPS groups should be compared, statistically, to baseline.

Minor points on this figure: Its not clear what is shown in 1a - the legend indicates it is an example of the microglial reconstruction but what 'traces' means is not clear.

1b - it is not clear that effects have peaked at 48 hours - the 48 hour example appears to show more complexity than the 24 hours and the 4 hours

Figure 3

The authors report that LPS increases plaque number in APP/PS1 but not in APP/PS1/NLRP3^{-/-}. Once again this is apparently based on a statistically significant post-hoc test in APP/PS1 and a failure to find this difference in the NLRP3^{-/-}. However, the magnitude of increase in plaques in the NLRP3^{-/-} is the same, if not greater than in APP/PS1, when examined on a proportional basis (that is, LPS doubles plaque number in NLRP3^{-/-} and increases them by slightly LESS in APP/PS1 with normal NLRP3 expression). I am really not sure it serves the data or the reader to present this as an increase in app/ps1 and no increase in NLRP3^{-/-}. Given that the authors have already published high profile papers on the role of NLRP3 in amyloid pathology per se, dissecting the effects of NLRP3 on disease per se, from effects of NLRP3 on LPS impacts on the same parameters are crucial in the current study.

It does look as though the effects of LPS on Ab1-40 and 1-42 are more prominent in the app/ps1 mice, but once again there are modest effects of LPS in NLRP3^{-/-} mice. However, the interpretation of the data as a whole should take into account the effects of LPS in NLRP3^{-/-}. I do not believe the effect is absent in those mice and I think the reader can see this, even in figure 3a.

Figure 4

The authors then turn their attention to phagocytic ability with respect to Amyloid beta after the LPS challenge.

It is relatively clear from the methoxy O4 FACS measures that microglia in APP/PS1 mice phagocytose less Ab in the days after LPS. Moreover the NLRP3 mice appear to phagocytose MORE. The statistics have not been fully detailed but the latter is not annotated as significant. I think this may be a robust and important result. According to the figure legend the authors performed a 2 way ANOVA of these data but do not report the results. An analysis with strain as one factor and LPS across time as a repeated measure would likely produce a main effect of LPS and perhaps a main effect of strain. However, I would expect that they may also see an interaction of strain and LPS over time, suggesting that LPS has opposite effects on phagocytosis depending on the strain. This analysis should be made explicit in the text (Full ANOVA for main effects and interactions, with F, df and p values) and the finding discussed appropriately (even if only a trend). This is important because it may not simply be the case that LPS impairs phagocytosis in a NLRP3-dependent way. If this is disease-state dependent then it has different implications.

Figure 7

Instead of citing unpublished observations on increased microglial proliferation in AD models the authors should simply cite published data supporting that statement (Olmos-Alonso et al., Brain, 2016). If it is the case that the data in figure do not refer at all to APP/PS1 mice this should be made explicit. The figures do indeed suggest that the control group is WT, but since the previous figures

have been with APP/PS1 it requires a mental gear-change to return to non-AD mice, so it would be helpful to the reader to emphasise this.

The Ki67 images are a little small for assessing the positive cells - often one can see the arrows but barely see the cells. The data seem to suggest that LPS produces an acute increase in proliferation and that up to 1 in 10 microglia can be seen to be dividing at 2 days post-LPS. Again the statistical analysis is not clear (what is inter group analysis ? - the authors never properly describe their statistical analysis and although such text can sometimes be cumbersome in the main text, it is important to describe clearly, somewhere, precisely what the analysis shows.

The main observations would appear to be that

- 1) proliferation is higher in the hippocampus than the cortex
- 2) in the hippocampus, proliferation is higher at 5 months than 15 months and
- 3) at 15 months proliferation is lowest in the NLRP3^{-/-} while at 5 months it appears higher in the NLRP3.

For 3) again it needs to be assessed explicitly whether there is an interaction between age and strain - because it looks possible that LPS has differential effects in NLRP3^{-/-} depending on age (increasing Ki67 in young but decreasing it in old). Can you provide the full statistical analysis?

Figure 8.

I found this text hard to follow: Here I break it down sentence by sentence to try to understand the statements made:

1: "In contrast, APP/PS1 and APP/PS1/Nlrp3^{-/-} mice already showed proliferating microglia in the cortex and hippocampus under control conditions mainly located at A β deposits (Fig 8a,b)": Compared to what? There are no non-transgenic animals in this figure.

2: "While there was no difference in the number of proliferating microglia between 5mo APP/PS1 and APP/PS1/NLRP3^{-/-} mice", : Looking at figure 8c there seems to be a clear difference between these 2 strains at 5 months since there are no proliferating microglia observed at "Con" in 5m APP/NLRP3^{-/-} while there are about 8% in "APP 5m".

3 "...NLRP3 knockout substantially reduced the number of inflammation-induced proliferating microglia in aged, 15mo animals (Fig 8c)": In both the cortex and the hippocampus LPS induces an acute increase (at 2d) in proliferating microglia and this increase looks quite robust to me in both APP 15m and APP/NLRP3^{-/-} mice, in both regions. The change from baseline (ie 2d minus Con) is very similar in both strains and if one was to calculate it as a fold increase from baseline (ie 2d over Con) then the fold increase would be even higher in the NLRP3 ko at 15 month. Once again, the DISEASE-ASSOCIATED proliferation appears to be reduced in NLRP3^{-/-} mice, but LPS is very well able to induce the increase in proliferation. I absolutely cannot see that knockout of NLRP3 blocks this proliferation as the authors suggest it does when they state that "NLRP3 knockout substantially reduced the number of inflammation-induced proliferating microglia"

The remaining figures (below) are fine, but a comment on the nature of cd169 labelling would be reasonable.

Figure 2. Age shows the expected increase in microglial #branches and path length but LPS fails to produce acute changes. These acute changes occur irrespective of age and the failure to change acutely fails to occur, irrespective of age. All looks good and appropriately analysed

Figure 5. Fine. Any changes seen in plaque distant microglia are very subtle.

Figure 6. CD169 positive cells, which should label peripheral monocytes but not microglia, were apparent only post-LPS, only in APP/PS1 mice at 15 months. The labelling for this marker looks rather punctate with puncta being considerably less than the size of cells

Minor changes:

In the Introduction "Cunningham, Wilcockson, Campion, Lunnon, 2005" looks like an unformatted citation and in the bibliography it is incom

Referee #2:

In this manuscript Tejera and colleagues investigated the effects of a single systemic LPS injection in young and aged mice in a mouse model of Alzheimer's disease and they describe these findings take place through a pathway involving NLRP3. Noteworthy, they used two-photon laser microscopy to characterize for the first time the morphological changes in vivo, hours and days after the LPS challenge using the CX3CR1-GFP transgenic mouse line. Furthermore, they imaged and assessed the changes occurring in eight different scenarios: comparing aged and young mice, wild type or knock-out for the *Nlrp3* gene, with and without APP/PS1 transgene.

Their main findings comprise that the changes in microglia morphology after LPS injection are influenced by age, the presence of App transgene, A β plaques and that they are somehow related to NLRP3. Additionally, they describe an increase in plaque number and a possible decrease in phagocytic activity in 15 months old APP/PS1 mice, the presence of cells expressing CD169 in the same aged group and study the effects of LPS administration on proliferation in the different groups. Even though the study is well performed and structured I have some major concerns about the overall novelty of this manuscript.

1. The study is not novel at all. Microglial retraction with the amount of LPS used is well known since at least two decades. The increase in plaque load was described by Wendeln AC et al. in Nature 2018, (which was not cited in the manuscript!). The authors seem to be aware of this lack of novelty and cited two more manuscripts in the discussion that assessed the increase of amyloid deposition in transgenic murine models of amyloidosis (Lee et al, 2008; Joshi et al, 2014).
2. The authors tried to give a mechanistic insight into the role of NLRP3 in microglia, but the data obtained questions the depth of this analysis. Altogether, the lack of novelty and lack of mechanistic insight raises major concerns in the referee for the eligibility of this manuscript for EMBO Journal.
3. *Nlrp3*^{-/-} microglia do not display morphological changes after the single LPS injection. Nevertheless, no mechanism is provided to explain how NLRP3 in microglial cells in the CNS parenchyma is activated by LPS. I would ask the authors to elucidate how the LPS is directly crossing the blood brain barrier and activating microglia through its specific receptor, or if there is an intermediate signal mediating this effect as suggested by other publications (Banks WA, et al. Brain Behav Immun. 2010 Jan. doi: 10.1016/j.bbi.2009.09.001).
4. CD169⁺ cells are assumed to be infiltrating peripheral myeloid cells. The authors should prove through fate mapping or any similar technique that the described CD169⁺ cells are indeed peripheral myeloid cells and not just microglia expressing this marker. Some authors claim this marker can be upregulated in microglia (Bogie JF, Mult Scler. 2018 Mar. doi: 10.1177/1352458517698759).
5. In Figure 4c, microglia cells are gated simply as living CD11b⁺ cells. A better gating strategy is required specially if you are claiming the possibility of infiltrating peripheral myeloid cells. After targeting leucocytes, discarding duplex cells and gating living cells with the Live/Dead staining; CD3, CD19 and GR1 cells must be excluded. Finally microglia cells must be gated not only by CD11b⁺ but CD45 low to distinguish from peripheral myeloid cells. Additionally, Figure 4c seems to display a subpopulation of cells in the upper right part of the APP/NLRP3^{-/-} FACS dot plot that is not present in the APP mice. If this population does not disappear after a proper gating strategy, the authors should describe what they are.
6. Please explain the relevance of your findings in Figure 4a,b and 5a,b given the distinct populations already described by Keren-Shaul H et al. Cell 2017 which seem to be the same as yours.
7. The authors claim LPS related changes act through the NLRP3 pathway but never show proof of NLRP3 being activated in their model. The study would first benefit from a proof of principle that LPS is directly or indirectly triggering NLRP3 inflammasome activation and therefore induces ASC-spec formation that would explain the changes observed. Please show as well quantitative data of Asc formation, especially in Figures 2 and 6c,d. Given the experience the author's group has with immunofluorescence detection of Asc-Specs, this should not be a problem.
8. The term "inflammation" is wrongly used throughout the whole manuscript (see: Aguzzi et al. Science. 2013 Jan 11;339(6116):156) and should be avoided. There is simply no inflammation during neurodegeneration such AD. There is a difference to MS, meningitis, stroke etc.

Minor Points:

1. It is better to represent each count as a dot instead of Bars to appreciate the distribution of the sample. It is best not to hide this information. You may overlay the dots on top of the bars. Also, the correct measure of error bars to show is Standard Deviation and not SEM. SEM is only for populations while SD talks about the variability within a sample.

2. The nomenclature for the Nlrp3 gene in mice must be written with only the first in capital and everything in cursive, whereas the NLRP3 protein in mice must be written all letters in capital. The same applies for other transgenes used. This was taken into account but inconsistent in the text and not at all taken into account in the figures. For references guidelines please read:
(<http://www.informatics.jax.org/mgihome/nomen/gene.shtml>,
https://en.wikipedia.org/wiki/Gene_nomenclature,
https://www.ncbi.nlm.nih.gov/genome/doc/internatprot_nomenguide/)
3. Related information is spread across too many figures. I suggest sending non-crucial information to more Extended Views and fuse into one figure the following: Figures 1 and 2. Figures 4 and 5. Figures 7 and 8.
4. It is odd that the Figure 1a is not mentioned in the main text but only in the methods. Please include in the main text or relocate it to an Extended View.
5. Figure 1e and Figure 2a displays CX3-eGFP above the timeline. It should say CX3CR1-eGFP.
6. Please quantify number of plaques and plaque size separating cortex and hippocampus for figure 3 (as done in Figures 7 and 8).
7. Please explain the meaning of the topmost significance bar in all the graphs of Figure 3b. Clarify if it corresponds to a significant difference between each bar and its equivalent in the Nlrp3 knock out group or if it's indicating significant differences between the first and last bar of both groups. The same applies to the bars in Figure 7c and EV1.
8. Figure 6 should include a quantification of the CD169+ cells described to assess the relevance of such findings. Please indicate in % of CD169+ cells from the total Iba-1 population surrounding the plaques and the % of plaques surrounded by CD169+ cells from total plaques in image.
9. The image quality of the Ki67 stainings in figures 7 & 8 is not optimal. Please include a higher magnification image where Ki67, DAPI and Iba1 can be evidently seen overlaid. The same for the CD169 staining in figure 6. It would be best if you could present the individual channels in high magnification and at the end an overlay of all channels.
10. In figures 7 and 8. DAPI is not indicated in the legend of each image.
11. In figure 8a&b, Dapi and Methoxy should not be used use in the same color, this creates confusion. Please repeat representative images using Thioflavine-S or Thiazine red that stains the same structures and move Ki67 to another channel like far-red.
12. In Figure 8c, please explain why Ki67 decreases in APP 5m old mice after LPS injection. This piece of data contradicts your general statement for that figure.

Referee #3:

The manuscript "Systemic inflammation impairs microglial Ab clearance through NLRP3 inflammasome" aims to investigate the effect of systemic inflammation on microglial activation with regard to aging, Alzheimer's Disease (AD) and the NLRP3 inflammasome. Microglia of mice expressing CX3CR1-GFP were monitored by Two-photon laser scanning microscopy in vivo at different time points after LPS injection. Microglia of 15 months old mice showed less ramification in general compared to microglia from 5 months old mice and animals of both age groups exhibited a more "activated" phenotype (reduced branching, reduced amount and length of processes) 2 d but not 10 d after LPS treatment. LPS treatment resulted also in an increased number of amyloid beta plaques and higher levels of Abeta 40 and 42. Lack of the inflammasome component NLRP3 (NLRP3 KO mice) prevented morphological changes in microglia (non-AD model and plaque-distant), Abeta increase and infiltration of CD169-positive peripheral macrophages. Furthermore, microglial proliferation was increased 2 d after LPS challenge in all mouse models and ages with less proliferation in the hippocampus of NLRP3 KO mice occurring.

The manuscript is well structured but some figures should be combined or moved to EV to avoid repetitions and also the writing style must be improved to increase the clarity of the text. The topic - dissecting the effect of systemic inflammation on microglia in the context of AD - is certainly of high interest and the reviewer appreciates the efforts to demonstrate these changes in an in vivo setting. However, there are a couple of major concerns, which must be addressed:

Major Points:

1. The authors need to demonstrate the existence and the time course of a systemic inflammation

after LPS injection by additional markers apart from microglia morphology. In addition to cytokines, (see 2) also staining of addtl. microglial activations makers (such as CD68 and Clec7a) is required.

2. The manuscript is mainly descriptive and would profit - in regard to novelty - from a deeper mechanistic insight. The authors found a protective effect from lack of NLRP3 at various levels. This indicates a major involvement of not only cytokines in general but also the IL1beta/IL18 pathway, specifically. Therefore, the authors must determine the cytokine levels before and after injection of LPS in the periphery and the brain a) to elucidate the protective effect of NLRP3 deletion in the aging paradigm (as discussed in regard to Fig 2b,c,d) as well as during systemic inflammation and b) whether release of TNFalpha and IL6 (also induced by LPS but not dependent on NLRP3) is indeed irrelevant in this setting as indirectly suggested by these data.

3. Soma size is another parameter of microglia activation (as the authors also mention in the discussion). Looking at Fig. 1b, this parameter does not seem to support the conclusion, namely that all activation parameters are at its maximum after 2 d. Are the pictures in Fig. 1b not representative? Please provide a quantification of this parameter.

4. While the effect of LPS on proliferation has been shown previously (e.g. Shankaran et al., 2007 DOI:10.1002/jnr.21389), the outcome of LPS stimulation on Abeta load is interesting (Fig. 3), where the authors propose reduced phagocytosis as possible mechanism (Fig. 4 c,d). As phagocytosis decreased progressively during the observation time (up to 10 d), a longer observation time is required - especially regarding future therapeutic inventions - to determine if a systemic inflammation has transient or a permanent effect.

5. Are the changes the authors observed in microglia reflected in astrocytes (or other CNS cells) or is this a microglia-specific effect?

6. The hypothesis on page 9 is pretty bold: "NLRP3 knockout influences infiltration of A β -directed migration at several levels including the reduction of inflammatory mediators, the capability of peripheral immune cells to enter the brain and also by protecting the integrity of the blood brain barrier". Since the authors only show a lack of infiltrating CD169 positive cells, this statement needs to be adjusted - or substantiated. The least the authors can do is to show cytokine levels (as inflammatory mediators) as suggested in 2, as well as convincing data regarding the integrity of the BBB (e.g. an Evans blue staining).

Minor Points:

1. Fig 2B 5 months 2nd row, 15 months 2nd row and Fig. 4A APP: are these really the same regions because the vascularization looks different.

2. Fig 4c,d: how was the phagocytosis assay performed, this lacks in the Methods section.

3. Fig 4a and 5a: please depict in the figure and the figure legend the antibodies that were used in their respective color.

4. Please detail what is considered 'in vicinity of plaques' as well as 'distant from plaques' for microglia. If this is based on the methoxyX04 staining, what about the plaque halo that wouldn't be visible with this beta-sheet staining?

5. The paragraph describing Figs 7 and 8 needs to be revised:

a. please change cortical to hippocampal (3rd line from the bottom)

b. Why is there no mentioning of the strong increase in cell proliferation 2 d after LPS treatment apart from the section title?

c. The authors state that "While there was no difference in the number of proliferating microglia between 5mo APP/PS1 and APP/PS1/NLRP3^{-/-} mice, NLRP3 knockout substantially reduced the number of inflammation-induced proliferating microglia in aged, 15mo animals (Fig 8c). According to Fig. 8c there is a significant difference also in 5 month old animals.

6. Fig. EV3: The blot mainly shows differences in Actin and not in Beclin1.

7. Page 9: "In AD and related mouse models, microglial proliferation seems to be accelerated (unpublished observations)." It seems the authors are referring to microgliosis, which is well documented in AD and should be referenced appropriately.

8. The statement on page 8: "we observed that these mice were largely refractory to peripheral immune challenge and aging, since no morphological changes were observed upon LPS challenge"

solely based on morphological changes is a bit farfetched (see also 1 and 2).
9. Please mention the exact location of the cranial window in the methods section.

1st Revision - authors' response

7th May 2019

Referee #1:

Briefly, LPS given peripherally changed microglial morphology as measured by 2 photon microscopy and made them less able to clear amyloid beta. That change is reported to be NLRP3-dependent.

KEY QUESTION: NLRP3^{-/-} mice are already protected against the APP/PS1 phenotype so we need to be sure that any beneficial effects of the KO are clearly shown to be selective to the LPS-induced changes and not simply explained by the different underlying condition at the time of the LPS challenge.

There are multiple interesting and potentially important observations here, but the key assertion that the effects of LPS on microglial activation, phagocytosis, proliferation and monocyte infiltration are not fully supported by the data as they currently stand. The data need more explicit statistical analysis (throughout) in order to increase confidence in some of the findings and in some cases the data simply do not support the conclusions. It is very important that the known affect of NLRP3^{-/-} on some of these parameters in APP/PS1 is not mistaken for a NLRP3-dependent effect on LPS on those same parameters.

Response: we thank the reviewer the suggestion. In the subsequent points we address the statistical concerns raised by the reviewer by providing detailed information about the statistical analysis. We also try to describe better the experimental conditions and the respective controls which will explain, which changes and which aspect of protection are mediated by LPS and NLRP3 knockout.

Results

Figure 1

1f - "recovery was restricted to normalization of the branch number". It is not clear what this means, since all parameters seem to recover at 10d back to control levels. It is not clear what statistical justification is used for this, but it is absolutely clear that they are not significantly different from their baseline level. The benefits of real-time and repeated measures allows the authors to see that there are changes at 2 days but by 10 days they are once again indistinguishable from controls: LPS-induced reductions in branch number, path length, and maximum branch order all recover to levels that are indiscernibly different from control levels. To say that these microglia are not different to 2d is entirely counterintuitive. To look at deflections from control levels both post-LPS groups should be compared, statistically, to baseline.

Response: we agree with the comment made by the reviewer. We rewrote the sentence in order to more accurately reflect the results (page 6, lines 143-145).

Minor points on this figure: Its not clear what is shown in 1a - the legend indicates it is an example of the microglial reconstruction but what 'traces' means is not clear.

Response: we apologize for not being clear in this description. "Traces" make reference to the automatic reconstruction of the microglia processes. This change was introduced in the figure legend of the revised manuscript and the figure was moved to Fig EV1.

1b - it is not clear that effects have peaked at 48 hours. The 48-hour example appears to show more complexity than the 24 hours and the 4 hours

Response: we agree with the reviewer and display a more representative reconstruction image (Fig 1a).

Figure 3

The authors report that LPS increases plaque number in APP/PS1 but not in APP/PS1/NLRP3^{-/-}. Once again this is apparently based on a statistically significant post-hoc test in APP/PS1 and a failure to find this difference in the NLRP3^{-/-}. However, the magnitude of increase in plaques in the NLRP3^{-/-} is the same, if not greater than in APP/PS1, when examined on a proportional basis (that is, LPS doubles plaque number in NLRP3^{-/-} and increases them by slightly LESS in APP/PS1 with normal NLRP3 expression). I am really not sure it serves the data or the reader to present this as an increase in app/ps1 and no increase in NLRP3^{-/-}. Given that the authors have already published high profile papers on the role of NLRP3 in amyloid pathology per se, dissecting the effects of NLRP3 on disease per se, from effects of NLRP3 on LPS impacts on the same parameters are crucial in the current study.

Response: we thank the reviewer the comment. Due to big dispersion in the data, we have increased the number of mice and separately analyzed cortex and hippocampus. Our results (included in the revised version of the manuscript) suggest an increased amyloid load, assessed by immunohistochemistry, in APP/PS1 but not in APP/*Nlrp3*^{-/-} for both, cortex and hippocampus (Fig 3a,b and EV7a,b).

It does look as though the effects of LPS on Ab1-40 and 1-42 are more prominent in the app/ps1 mice, but once again there are modest effects of LPS in NLRP3^{-/-} mice. However, the interpretation of the data as a whole should take into account the effects of LPS in NLRP3^{-/-}. I do not believe the effect is absent in those mice and I think the reader can see this, even in figure 3a.

Response: we understand the reviewer's criticism, however, statistics were re-checked. Two-way ANOVA was used and Tukey for multiple comparisons was used as a post-hoc test. The interaction resulted is a non-significant result with a p-value of 0.1324 and the F (6, 26) = 1.829 for Ab 1-40. For Ab 1-42 the p-value was 0.0914 and the F (6, 28) = 2.054.

Figure 4

The authors then turn their attention to phagocytic ability with respect to Amyloid beta after the LPS challenge.

It is relatively clear from the methoxy O4 FACS measures that microglia in APP/PS1 mice phagocytose less Ab in the days after LPS. Moreover the NLRP3 mice appear to phagocytose MORE. The statistics have not been fully detailed but the latter is not annotated as significant. I think this may be a robust and important result. According to the figure legend the authors performed a 2 way ANOVA of these data but do not report the results. An analysis with strain as one factor and LPS across time as a repeated measure would likely produce a main effect of LPS and perhaps a main effect of strain. However, I would expect that they may also see an interaction of strain and LPS over time, suggesting that LPS has opposite effects on phagocytosis depending on the strain. This analysis should be made explicit in the text (Full ANOVA for main effects and interactions, with F, df and p values) and the finding discussed appropriately (even if only a trend). This is important because it may not simply be the case that LPS impairs phagocytosis in a NLRP3-dependent way. If this is disease-state dependent then it has different implications.

Response: we thank the reviewer for the comment. We re-checked our statistical analysis and indeed, found a significant interaction with a p-value of 0.0145 $F(2, 13) = 6,437$. However, after running Tukey post hoc test for multiple comparisons there was no significant difference between controls, 2 days or 10 days for the APP*Nlrp3*^{-/-} group.

Figure 7

Instead of citing unpublished observations on increased microglial proliferation in AD models the authors should simply cite published data supporting that statement (Olmos-Alonso et al., Brain, 2016). If it is the case that the data in figure do not refer at all to APP/PS1 mice this should be made explicit. The figures do indeed suggest that the control group is WT, but since the previous figures have been with APP/PS1 it requires a mental gear-change to return to non-AD mice, so it would be helpful to the reader to emphasize this.

Response: the reference suggested by the reviewer was added to the revised manuscript. Figure 7 makes reference to non-APP mice. That means WT and *Nlrp3*^{-/-} PBS and LPS treated. We have incorporated this change into the revised version of the manuscript (page 11, lines 296-297).

The Ki67 images are a little small for assessing the positive cells - often one can see the arrows but barely see the cells. The data seem to suggest that LPS produces an acute increase in proliferation and that up to 1 in 10 microglia can be seen to be dividing at 2 days post-LPS. Again the statistical analysis is not clear (what is inter group analysis? - the authors never properly describe their statistical analysis and although such text can sometimes be cumbersome in the main text, it is important to describe clearly, somewhere, precisely what the analysis shows.

Response: we thank the reviewer for the comment; the images were enlarged so that the reader can now clearly see the cells and appreciate the findings. Inter-group analysis makes reference to WT versus *Nlrp3*^{-/-} comparison considering the three different time-points. The reviewer is right, the

expression could lead to confusion and was removed in the revised version of the manuscript.

The main observations would appear to be that

- 1) proliferation is higher in the hippocampus than the cortex
- 2) in the hippocampus, proliferation is higher at 5 months than 15 months and
- 3) at 15 months proliferation is lowest in the NLRP3^{-/-} while at 5 months it appears higher in the NLRP3.

For 3) again it needs to be assessed explicitly whether there is an interaction between age and strain - because it looks possible that LPS has differential effects in NLRP3^{-/-} depending on age (increasing Ki67 in young but decreasing it in old). Can you provide the full statistical analysis?

Response: we agree with the reviewer comment. Microglia proliferation in *Nlrp3*^{-/-} at the age of 5 months in the hippocampus is higher than in 15 months (With a p value of 0.0009, Tukey for multiple comparison test). We have now added these points (page 11 lines 304-307) and also provide a full statistical analysis of the data.

Figure 8.

I found this text hard to follow: Here I break it down sentence by sentence to try to understand the statements made:

1: "In contrast, APP/PS1 and APP/PS1/*Nlrp3*^{-/-} mice already showed proliferating microglia in the cortex and hippocampus under control conditions mainly located at A β deposits (Fig 8a,b)": Compared to what? There are no non-transgenic animals in this figure.

Response: we thank the reviewer the comment. This is compared to the observation done in figure 7 where in PBS-treated mice we did not observe proliferative microglia (Fig 7). However, in figure 8, proliferative microglia was found in PBS-treated mice. This sentence has been added to the revised version of the manuscript (Page 11, line 309).

2: "While there was no difference in the number of proliferating microglia between 5mo APP/PS1 and APP/PS1/NLRP3^{-/-} mice": Looking at figure 8c there seems to be a clear difference between these 2 strains at 5 months since there are no proliferating microglia observed at "Con" in 5m APP/NLRP3^{-/-} while there are about 8% in "APP 5m".

Response: the reviewer is right. The proliferation is higher in the cortex of APP/PS1 compared to APP/NLRP3^{-/-}. This observation has been included in the revised version of the manuscript (page 12 line 310-311).

3 "...NLRP3 knockout substantially reduced the number of inflammation-induced proliferating microglia in aged, 15mo animals (Fig 8c)": In both the cortex and the hippocampus LPS induces an acute increase (at 2d) in proliferating microglia and this increase looks quite robust to me in both APP 15m and APP/NLRP3^{-/-} mice, in both regions. The change from baseline (ie 2d minus Con) is very similar in both strains and if one was to calculate it as a fold increase from baseline (ie 2d over Con) then the fold increase would be even higher in the NLRP3 ko at 15 month. Once again, the

DISEASE-ASSOCIATED proliferation appears to be reduced in NLRP3^{-/-} mice, but LPS is very well able to induce the increase in proliferation. I absolutely cannot see that knockout of NLRP3 blocks this proliferation as the authors suggest it does when they state that "NLRP3 knockout substantially reduced the number of inflammation-induced proliferating microglia"

Response: we agree with the reviewer. In APP/PS1 and APP/PS1/*Nlrp3* mice, LPS is equally stimulating microglia proliferation. We corrected this in revised version of the manuscript (page 12 line 313).

The remaining figures (below) are fine, but a comment on the nature of cd169 labeling would be reasonable.

Response: A comment on the nature of CD169 has been added to the revised version of the manuscript (page 10 lines 264-266).

CD 169 (*Siglec 1*) has been described as a marker of inflammatory monocytes and differentiated dendritic cells and it has been reported to discriminate microglia from infiltrating myeloid cells (Perez *et al*, 2017; Butovsky *et al*, 2012; Rice *et al*, 2015). Nevertheless we cannot exclude an upregulation of this marker by microglia under the given experimental conditions. Even though our results (included in the revised version of the manuscript) suggest that BBB is compromised in APP 15mo mice after LPS injection (see Fig EV12). Such a disruption of the BBB could have been prompted the infiltration of peripheral cells given the presence of cytokines and chemokines in the brains of APP/PS1 mice

Figure 2. Age shows the expected increase in microglial #branches and path length but LPS fails to produce acute changes. These acute changes occur irrespective of age and the failure to change acutely fails to occur, irrespective of age. All looks good and appropriately analyzed

Figure 5. Fine. Any changes seen in plaque distant microglia are very subtle.

Figure 6. CD169 positive cells, which should label peripheral monocytes but not microglia, were apparent only post-LPS, only in APP/PS1 mice at 15 months. The labeling for this marker looks rather punctate with puncta being considerably less than the size of cells

Response: we thank the reviewer for the comment. Images were taken again and improved for proper visualization (Fig 6 and EV11).

Minor changes:

In the Introduction "Cunningham, Wilcockson, Campion, Lunnon, 2005" looks like an unformatted citation and in the bibliography it is incomplete

Response: reference was corrected for the revised version of the manuscript.

Referee #2:

In this manuscript Tejera and colleagues investigated the effects of a single systemic LPS injection in young and aged mice in a mouse model of Alzheimer's disease and they describe these findings take place through a pathway involving NLRP3. Noteworthy, they used two-photon laser microscopy to characterize for the first time the morphological changes *in vivo*, hours and days after the LPS challenge using the CX3CR1-GFP transgenic mouse line. Furthermore, they imaged and assessed the changes occurring in eight different scenarios: comparing aged and young mice, wild type or knock-out for the *Nlrp3* gene, with and without APP/PS1 transgene.

Their main findings comprise that the changes in microglia morphology after LPS injection are influenced by age, the presence of App transgene, A β plaques and that they are somehow related to NLRP3. Additionally, they describe an increase in plaque number and a possible decrease in phagocytic activity in 15 months old APP/PS1 mice, the presence of cells expressing CD169 in the same aged group and study the effects of LPS administration on proliferation in the different groups.

Even though the study is well performed and structured I have some major concerns about the overall novelty of this manuscript.

1. The study is not novel at all. Microglial retraction with the amount of LPS used is well known since at least two decades. The increase in plaque load was described by Wendeln AC et al. in Nature 2018, (which was not cited in the manuscript!). The authors seem to be aware of this lack of novelty and cited two more manuscripts in the discussion that assessed the increase of amyloid deposition in transgenic murine models of amyloidosis (Lee et al, 2008; Joshi et al, 2014).

Response: We apologize for not originally citing the paper by Wendeln et al., which now has been included. Regarding the lack of novelty in our paper claimed by the reviewer, we would like to emphasize that the novelty of our manuscript lies in showing the effect of *Nlrp3* in LPS-induced Abeta pathology (Fig 3 and EV7) which to the best of our knowledge has not been described before. Moreover, we showed in the manuscript the *in vivo* changes in individual microglia cells by longitudinal two-photon microscopy (Fig 1, 2, 4 and 5).

Additionally, we suggest that the increase in Abeta pathology is due to a decrease in its clearance (Fig 4c,d and EV10). Regarding the study published by Wendeln et al., we need to mention that the experimental paradigm is different (4 LPS injection in Wendeln et al). The time that amyloid pathology was evaluated after last LPS injection was 6 months, which converts this in a long-term evaluation. Concomitantly, a deep transcriptomic analysis was performed showing changes in microglia profiles but no mechanism whatsoever was provided for the increase in amyloid pathology.

2. The authors tried to give a mechanistic insight into the role of NLRP3 in microglia, but the data obtained questions the depth of this analysis. Altogether, the lack of novelty and lack of mechanistic insight raises major concerns in the referee for the eligibility of this manuscript for EMBO Journal.

Response: we consider there are novel aspects in the manuscript such as the involvement of the NLRP3 inflammasome for changes of Abeta pathology induced by systemic inflammation. Most importantly, we longitudinally assess changes of microglia dynamics *in vivo*. We believe that we

now provide more mechanistic information including on the inflammatory response in brain and periphery (Fig EV5), BBB disruption (Fig EV12) and the inflammasome activation (Fig EV4).

3. *Nlrp3*^{-/-} microglia do not display morphological changes after the single LPS injection. Nevertheless, no mechanism is provided to explain how NLRP3 in microglial cells in the CNS parenchyma is activated by LPS. I would ask the authors to elucidate how the LPS is directly crossing the blood brain barrier and activating microglia through its specific receptor, or if there is an intermediate signal mediating this effect as suggested by other publications (Banks WA, et al. *Brain Behav Immun.* 2010 Jan. doi: 10.1016/j.bbi.2009.09.001).

Response: We thank the reviewer for the suggestion. We have measured IL-1beta and TNF-alpha levels in the liver and the brain in order to dissect both, the peripheral and central effects of LPS challenge in the inflammatory response (Fig EV5). We have found considerable less IL-1beta levels after LPS injection in *Nlrp3*^{-/-} mice either in the liver or brain. Therefore, one plausible explanation on how *Nlrp3*^{-/-} mice are refractory to peripheral LPS could be the overall reduction IL-1beta-driven inflammatory response (Fig EV5). These results were included in the results section of the revised manuscript (page 7, lines 177-180). Regarding LPS extravasation, we thank the reviewer for the suggested paper. The results shown there are for intravenously LPS injection.

We have checked and to the best of our knowledge, there is no publication showing LPS extravasation after an intraperitoneally injection (Hoogland *et al*, 2015).

4. CD169⁺ cells are assumed to be infiltrating peripheral myeloid cells. The authors should prove through fate mapping or any similar technique that the described CD169⁺ cells are indeed peripheral myeloid cells and not just microglia expressing this marker. Some authors claim this marker can be upregulated in microglia (Bogie JF, *Mult Scler.* 2018 Mar. doi: 10.1177/1352458517698759).

Response: We thank the reviewer for the comment. The paper suggested by the reviewer shows CD169⁺/Iba⁺ cells in an EAE murine model. This is a model where blood brain barrier disruption has been shown (Davalos *et al*, 2012), which makes difficult the discrimination between brain-resident cells and infiltrating ones. Based on this, we wanted to verify blood-brain barrier integrity and we have included for the revised version of the manuscript, fibrinogen staining (Fig EV12). Fibrinogen is abundant in the blood and participates in the coagulation cascade creating fibrin. Particularly in the nervous system, fibrinogen is deposited as fibrin once the blood brain barrier is disrupted (Mendiola *et al*, 2017).

Our results show the presence of fibrinogen deposits 2 and 10 days after LPS injection in APP 15mo mice (Fig EV12b,c). Importantly, no fibrinogen deposition was observed in APP/*Nlrp3*^{-/-} 15mo injected with LPS. These results suggest that peripheral LPS challenge disrupts blood-brain barrier in 15mo APP mice, which could lead to the infiltration of peripheral cells. We included these results into the result section (page 11, lines 275-284).

5. In Figure 4c, microglia cells are gated simply as living CD11b⁺ cells. A better gating strategy is required specially if you are claiming the possibility of infiltrating peripheral myeloid cells. After

targeting leucocytes, discarding duplex cells and gating living cells with the Live/Dead staining; CD3, CD19 and GR1 cells must be excluded. Finally microglia cells must be gated not only by CD11b+ but CD45 low to distinguish from peripheral myeloid cells. Additionally, Figure 4c seems to display a subpopulation of cells in the upper right part of the APP/NLRP3^{-/-} FACS dot plot that is not present in the APP mice. If this population does not disappear after a proper gating strategy, the authors should describe what they are.

Response: We thank the reviewer for the comment. We have added our gating strategy for the revised version of the manuscript (Fig EV9). For our gating strategy we first removed doublets cells using the FSC parameters, next dead cells were removed using a live/dead dye. For our methoxy analysis we took cells that were positive for both, CD11b and CD45. While the reviewer described the gating strategy to eliminate monocyte populations based on CD45 expression, this is only relevant in the healthy adult brain. In APP/PS1 mice it has been described that microglia particularly close to the plaque upregulate CD45 (Maier *et al*, 2008; Keren-shaul *et al*, 2017). Since these are the main cells of interest, we were reluctant to exclude cells based on CD45 expression. Cells expressing CD45 and not CD11b were excluded, as these could be T and B cells. We are completely aware of the difficulties using these settings to decipher between infiltrating myeloid cells and microglia and therefore without adequate markers available to accurately distinguish these populations in AD, we assess all myeloid cells instead.

6. Please explain the relevance of your findings in Figure 4a,b and 5a,b given the distinct populations already described by Keren-Shaul H *et al*. Cell 2017, which seem to be the same as yours.

Response: we thank the reviewer for the comment. We would like to acknowledge the lack of novelty in terms of the population heterogeneity. However, our manuscript shows for the first time in vivo dynamics of microglia in the context of systemic inflammation in AD. Importantly, we show in our manuscript that the populations react in different ways to LPS challenge (Fig 4a,b, 5a,b). Additionally, considering these populations are morphologically different (Fig EV8) we cannot analyze them together.

7. The authors claim LPS related changes act through the NLRP3 pathway but never show proof of NLRP3 being activated in their model. The study would first benefit from a proof of principle that LPS is directly or indirectly triggering NLRP3 inflammasome activation and therefore induces ASC-spec formation that would explain the changes observed. Please show as well quantitative data of Asc formation, especially in Figures 2 and 6c,d. Given the experience the author's group has with immunofluorescence detection of Asc-Specs, this should not be a problem.

Response: we have included the figure showing ASC/Iba-1 staining and quantification for the revised version of the manuscript. Our results indicate that peripheral immune challenge trigger ASC speck formation in a *Nlrp3*-dependent manner (Fig EV4). Interestingly, 10 days after LPS injection, we observed a reduction in the number of specks compared to 2 days post LPS (Fig EV4).

In summary, these results (together with IL-1beta levels) confirm the activation of the NLRP3 inflammasome.

8. The term "inflammation" is wrongly used throughout the whole manuscript (see: Aguzzi et al. Science. 2013 Jan 11;339(6116):156) and should be avoided. There is simply no inflammation during neurodegeneration such AD. There is a difference to MS, meningitis, stroke etc.

Response: we use throughout the manuscript the expression "systemic inflammation" which makes reference to our LPS peripheral stimulation. For the Central Nervous System we use the word neuroinflammation. We thank the reviewer for referencing the exquisite review by Ben Barres and colleagues. Here we transcribe a segment from the same review "Whereas viral, bacterial, and autoimmune diseases of the CNS can resemble their extraneural counterparts morphologically, the concept of "neuroinflammation" has gradually expanded to also describe diseases that display none of Celsus' cardinal signs". Of course we are aware that the inflammatory processes taking place during AD they may not present all of Celsus signs, however there is no place in the review referenced by the reviewer where states that no inflammation is taking place during AD.

Minor Points:

1. It is better to represent each count as a dot instead of Bars to appreciate the distribution of the sample. It is best not to hide this information. You may overlay the dots on top of the bars. Also, the correct measure of error bars to show is Standard Deviation and not SEM. SEM is only for populations while SD talks about the variability within a sample.

Response: we have checked EMBO Journal papers and graph are represented with SEM

2. The nomenclature for the Nlrp3 gene in mice must be written with only the first in capital and everything in cursive, whereas the NLRP3 protein in mice must be written all letters in capital. The same applies for other transgenes used. This was taken into account but inconsistent in the text and not at all taken into account in the figures. For references guidelines please read:

(<http://www.informatics.jax.org/mgihome/nomen/gene.shtml>, https://en.wikipedia.org/wiki/Gene_nomenclature, https://www.ncbi.nlm.nih.gov/genome/doc/internatprot_nomenguide/)

Response: we thank the reviewer for the observation. We apologize and corrected the nomenclature all through the revised version of the manuscript.

3. Related information is spread across too many figures. I suggest sending non-crucial information to more Extended Views and fuse into one figure the following: Figures 1 and 2. Figures 4 and 5. Figures 7 and 8.

Response: we thank the reviewer the suggestion. We have moved some non-crucial information to extended view as suggested.

4. It is odd that the Figure 1a is not mentioned in the main text but only in the methods. Please include in the main text or relocate it to an Extended View.

Response: we thank the reviewer the comment. We have moved figure 1a to an extended view (Fig EV1).

5. Figure 1e and Figure 2a displays CX3-eGFP above the timeline. It should say CX3CR1-eGFP.

Response: we have corrected the nomenclature.

6. Please quantify number of plaques and plaque size separating cortex and hippocampus for figure 3 (as done in Figures 7a,b and 8).

Response: We thank the reviewer for this suggestion; we have now quantified cortex and hippocampus separately (Fig 3 and EV7).

7. Please explain the meaning of the topmost significance bar in all the graphs of Figure 3b. Clarify if it corresponds to a significant difference between each bar and its equivalent in the Nlrp3 knock out group or if it's indicating significant differences between the first and last bar of both groups. The same applies to the bars in Figure 7c and EV1.

Response: the significance bar at the top makes reference to the increased amount of amyloid deposits in the APP/PS1 mice in comparison to APP/PS1*Nlrp3*^{-/-} for all the time points.

8. Figure 6 should include a quantification of the CD169+ cells described to assess the relevance of such findings. Please indicate in % of CD169+ cells from the total Iba-1 population surrounding the plaques and the % of plaques surrounded by CD169+ cells from total plaques in image.

Response: we would like to emphasize that this result is qualitative and not quantitative. We wanted to show the presence of these cells, presumably of peripheral origin.

9. The image quality of the Ki67 stainings in figures 7 & 8 is not optimal. Please include a higher magnification image where Ki67, DAPI and Iba1 can be evidently seen overlaid. The same for the CD169 staining in figure 6. It would be best if you could present the individual channels in high magnification and at the end an overlay of all channels.

Response: we thank the reviewer for the comment; the images were enlarged so that the reader may better appreciate the findings. Regarding the representation using individual channels, we were concerned about showing the three channels for all 24 conditions measured. This would lead to 72 images per figure and would make it difficult for the reader to appreciate the major findings, we were trying to emphasize. Therefore, we tried instead to show images that more accurately reflected the key findings of the experiment.

10. In figures 7 and 8. DAPI is not indicated in the legend of each image.

Response: We thank the reviewer correction. We have incorporated this to the revised version of the manuscript.

11. In figure 8a&b, Dapi and Methoxy should not be used use in the same color, this creates confusion. Please repeat representative images using Thioflavine-S or Thiazine red that stains the same structures and move Ki67 to another channel like far-red.

Response: we appreciate the comment made by the reviewer. For the revised version of the manuscript we have increased the magnification of the images and we hope this do not create confusion anymore.

12. In Figure 8c, please explain why Ki67 decreases in APP 5m old mice after LPS injection. This piece of data contradicts your general statement for that figure.

Response: We agree with the reviewer, however, it seems that this phenomenon is observed only in the cortex but not in the hippocampus. We suggest that – as most KI67 positive microglia are found at sites of amyloid deposition, the low number of such depositions may have lead to caused this variation. A cautionary note on this finding has been added to the text (page 12, lines 314-318).

Referee #3:

The manuscript "Systemic inflammation impairs microglial Ab clearance through NLRP3 inflammasome" aims to investigate the effect of systemic inflammation on microglial activation with regard to aging, Alzheimer's Disease (AD) and the NLRP3 inflammasome. Microglia of mice expressing CX3CR1-GFP were monitored by Two-photon laser scanning microscopy in vivo at different time points after LPS injection. Microglia of 15 months old mice showed less ramification in general compared to microglia from 5 months old mice and animals of both age groups exhibited a more "activated" phenotype (reduced branching, reduced amount and length of processes) 2 d but not 10 d after LPS treatment. LPS treatment resulted also in an increased number of amyloid beta plaques and higher levels of Abeta 40 and 42. Lack of the inflammasome component NLRP3 (NLRP3 KO mice) prevented morphological changes in microglia (non-AD model and plaque-distant), Abeta increase and infiltration of CD169-positive peripheral macrophages. Furthermore, microglial proliferation was increased 2 d after LPS challenge in all mouse models and ages with less proliferation in the hippocampus of NLRP3 KO mice occurring.

The manuscript is well structured but some figures should be combined or moved to EV to avoid repetitions and also the writing style must be improved to increase the clarity of the text. The topic - dissecting the effect of systemic inflammation on microglia in the context of AD - is certainly of high interest and the reviewer appreciates the efforts to demonstrate these changes in an in vivo setting. However, there are a couple of major concerns, which must be addressed:

Major Points:

1. The authors need to demonstrate the existence and the time course of a systemic inflammation after LPS injection by additional markers apart from microglia morphology. In addition to

cytokines, (see 2) also staining of addtl. microglial activations makers (such as CD68 and Clec7a) is required.

Response: we thank the reviewer for the comment. We have now added CD68 staining into the revised version of the manuscript (Fig EV3). Our results suggest a *Nlrp3*-dependent increase in CD68 immunoreactivity 2 days after LPS injection in 15mo WT mice. Interestingly, and in concordance with our morphological analysis, 10 days after peripheral immune challenge we found a reduction in CD68 immunoreactivity. We have included these results into the revised version of the manuscript (page 6, lines 145-150 and 7, lines 164-166).

2. The manuscript is mainly descriptive and would profit - in regard to novelty - from a deeper mechanistic insight. The authors found a protective effect from lack of NLRP3 at various levels. This indicates a major involvement of not only cytokines in general but also the IL1beta/IL18 pathway, specifically. Therefore, the authors must determine the cytokine levels before and after injection of LPS in the periphery and the brain a) to elucidate the protective effect of NLRP3 deletion in the aging paradigm (as discussed in regard to Fig 2b,c,d) as well as during systemic inflammation and b) whether release of TNFalpha and IL6 (also induced by LPS but not dependent on NLRP3) is indeed irrelevant in this setting as indirectly suggested by these data.

Response: we thank the input from the reviewer. Cytokine data was added to the revised version of the manuscript (Fig EV5) (page7, lines 177-180). Our results indicate for both liver and brain an increase in IL-1beta (*Nlrp3*-dependent) 2 days after LPS injection. Importantly, 10 days after LPS injection IL-1beta levels were the same as PBS-treated mice. Moreover, in order to get deeper knowledge on inflammasome activation, we have included to the revised version of the manuscript ASC staining (Fig EV4). Our results show that LPS peripheral challenge induces ASC speck formation by microglia.

Additionally, TNF-alpha levels showed an *Nlrp3*-independent increase 2 days post-LPS and then, 10 days later a decrease, showing similar levels to PBS-treated mice (Fig EV5).

3. Soma size is another parameter of microglia activation (as the authors also mention in the discussion). Looking at Fig. 1b, this parameter does not seem to support the conclusion, namely that all activation parameters are at its maximum after 2 d. Are the pictures in Fig. 1b not representative? Please provide a quantification of this parameter.

Response: the reviewer was correct in this observation. We have changed the images in the figure to more accurately reflect the findings described i.e where it is clear there are changes in soma size. It is important to mention also that increase in soma size is not a *sine qua non* condition in the microglia morphological changes (Gyoneva *et al*, 2014).

4. While the effect of LPS on proliferation has been shown previously (e.g. Shankaran *et al.*, 2007 DOI:10.1002/jnr.21389), the outcome of LPS stimulation on Abeta load is interesting (Fig. 3), where the authors propose reduced phagocytosis as possible mechanism (Fig. 4 c,d). As phagocytosis decreased progressively during the observation time (up to 10 d), a longer observation

time is required - especially regarding future therapeutic inventions - to determine if a systemic inflammation has transient or a permanent effect.

Response: We agree with the reviewer that principally, it would be very interesting to assess the longitudinal consequences of a single or multiple LPS challenges on microglial clearance capacity and beta-amyloid load. Nevertheless, we feel that this substantially exceeds the scope of the present study and would probably require longer intervals between observations (e.g. months instead of days), furthermore, such a study most likely would have to be performed in non-anesthetized animals to avoid the effects of repeated exposure to anesthetics and therefore could not directly be linked with the present study.

5. Are the changes the authors observed in microglia reflected in astrocytes (or other CNS cells) or are this microglia-specific effect?

Response: we thank the reviewer for the question. There is a recent paper indicating that reactive astrocytes require a microglial inflammatory response to become activated (Liddelow *et al*, 2017). Taking this into account we have now analyzed the astrocytic response to peripheral immune challenge and added these data to the revised manuscript (page 7, lines 188-200). Our results suggest that 2 days after LPS injection astrocytes become activated, increasing GFAP immunoreactivity (Fig EV6). Interestingly, *Nlrp3*^{-/-} mice did not exhibit changes in GFAP immunoreactivity corroborating that astrocytes require activated microglia in order to become reactive.

6. The hypothesis on page 9 is pretty bold: "NLRP3 knockout influences infiltration of A β -directed migration at several levels including the reduction of inflammatory mediators, the capability of peripheral immune cells to enter the brain and also by protecting the integrity of the blood brain barrier". Since the authors only show a lack of infiltrating CD169 positive cells, this statement needs to be adjusted - or substantiated. The least the authors can do is to show cytokine levels (as inflammatory mediators) as suggested in 2, as well as convincing data regarding the integrity of the BBB (e.g. an Evans blue staining).

Response: we thank the reviewer for this critique. In addition to the cytokine data, we have added fibrinogen staining as a measure of BBB disruption (Davalos *et al*, 2012) to the revised version of the manuscript (page 11 lines 275-284). As a consequence of BBB disruption fibrinogen forms deposits in the brain parenchyma. We have found fibrinogen deposits in APP 15mo 2 and 10 days post-LPS challenge (Figure EV12). Importantly, fibrinogen deposition resulted to be *Nlrp3*-dependent, since virtually no fibrinogen was found in APP/*Nlrp3*^{-/-} mice upon LPS injection. Altogether, we suggest that the inflammatory response that is taking place in both periphery and brain (Fig EV5) could promote BBB disruption and therefore cell infiltration of peripheral origin. Nevertheless, we have adjusted our interpretation to be more specific and also indicate that future studies will have to elucidate the precise contribution of peripheral cells (page 11, lines 290-291).

Minor Points:

1. Fig 2B 5 months 2nd row, 15 months 2nd row and Fig. 4A APP: are these really the same regions because the vascularization looks different.

Response: we thank the reviewer for the comment. The locations are indeed the same; the difference lies in the absorption of Dextran Red to stain the vasculature.

2. Fig 4c,d: how was the phagocytosis assay performed, this lacks in the Methods section.

Response: we apologize for the lack of a precise description. We have included a more detailed description to the revised version of the manuscript (page 22 lines 578-583). Analysis was performed as previously described (Kummer *et al*, 2012; Heneka *et al*, 2013).

3. Fig 4a and 5a: please depict in the figure and the figure legend the antibodies that were used in their respective color.

Response: Figures 4a and 5a were generated using two-photon microscopy, therefore no antibodies were used.

4. Please detail what is considered 'in vicinity of plaques' as well as 'distant from plaques' for microglia. If this is based on the methoxyX04 staining, what about the plaque halo that wouldn't be visible with this beta-sheet staining?

Response: we apologize for the lack of precision; "Vicinity of plaques" is defined as a region that is within 60µm of the plaque core. Indeed the *in vivo* labeling with methoxy-x04 will only show the core of the plaque and we have now highlighted this fact for a better understanding in the text (page 9, lines 232-233).

5. The paragraph describing Figs 7 and 8 needs to be revised:

a. please change cortical to hippocampal (3rd line from the bottom)

b. Why is there no mentioning of the strong increase in cell proliferation 2 d after LPS treatment apart from the section title?

c. The authors state that "While there was no difference in the number of proliferating microglia between 5mo APP/PS1 and APP/PS1/NLRP3^{-/-} mice, NLRP3 knockout substantially reduced the number of inflammation-induced proliferating microglia in aged, 15mo animals (Fig 8c). According to Fig. 8c there is a significant difference also in 5 month old animals.

Response: we thank the reviewer for the suggestions. All changes have been introduced to the revised version of the manuscript (page 12 lines 303-306).

6. Fig. EV3: The blot mainly shows differences in Actin and not in Beclin1.

Response: we thank the reviewer for the criticism. We have updated the figure with a more representative image of the Actin and Beclin 1 for the revised version of the manuscript (Fig EV10).

7. Page 9: "In AD and related mouse models, microglial proliferation seems to be accelerated (unpublished observations)." It seems the authors are referring to microgliosis, which is well documented in AD and should be referenced appropriately.

Response: we thank the reviewer for the suggestion. We have properly referenced (Olmos-Alonso *et al.*, 2016) the observations for the revised version of the manuscript (page 11, line 296).

8. The statement on page 8: "we observed that these mice were largely refractory to peripheral immune challenge and aging, since no morphological changes were observed upon LPS challenge" solely based on morphological changes is a bit farfetched (see also 1 and 2).

Response: we thank the reviewer for the comment. We have added cytokine data and CD68 staining as suggested by the reviewer before, to substantiate our morphological analysis. Unlike wild-type mice, *Nlrp3*^{-/-} mice did not show any increase in CD68 immunoreactivity upon LPS challenge (Fig EV3). In line with this result, no increase in IL-1beta was observed in *Nlrp3*^{-/-} mice after LPS injection (Fig EV5). These results in addition to the absence of ASC specks formation (Fig EV4) suggest that indeed, *Nlrp3*^{-/-} are relatively refractory to peripheral immune challenge. Nevertheless and taking into account that other markers could have been affected, we now rephrased the respective sentence to "we observed that these mice were largely refractory to peripheral immune challenge and aging, at least with respect to the evaluated markers of morphological change" (page 15, lines 404-406).

9. Please mention the exact location of the cranial window in the methods section.

Response: exact location of the cranial window has been added to the revised version of the manuscript (page 18, lines 466-467).

Reference:

- Butovsky O, Cudkowicz ME, Weiner HL, Murugaiyan G, Doykan CE, Wu PM, Gali RR & Iyer LK (2012) Modulating inflammatory monocytes with a unique microRNA gene signature ameliorates murine ALS Find the latest version : Modulating inflammatory monocytes with a unique microRNA gene signature ameliorates murine ALS. *J. Clin. Invest.* **122**: 3063–3087
- Davalos D, Ryu JK, Merlini M, Baeten KM, Le Moan N, Petersen M a, Deerinck TJ, Smirnov DS, Bedard C, Hakoziaki H, Gonias Murray S, Ling JB, Lassmann H, Degen JL, Ellisman MH & Akassoglou K (2012) Fibrinogen-induced perivascular microglial clustering is required for the development of axonal damage in neuroinflammation. *Nat. Commun.* **3**: 1227
- Gyoneva S, Davalos D, Biswas D, Swanger S a, Garnier-Amblard E, Loth F, Akassoglou K & Traynelis SF (2014) Systemic inflammation regulates microglial responses to tissue damage in vivo. *Glia* **62**: 1345–1360
- Heneka MT, Kummer MP, Stutz A, Delekate A, Schwartz S, Vieira-Saecker A, Griep A, Axt D, Remus A, Tzeng T-C, Gelpi E, Halle A, Korte M, Latz E & Golenbock DT (2013) NLRP3 is

- activated in Alzheimer's disease and contributes to pathology in APP/PS1 mice. *Nature* **493**: 674–8
- Hoogland ICM, Houbolt C, Westerloo DJ Van, Gool WA Van & Beek D Van De (2015) Systemic inflammation and microglial activation : systematic review of animal experiments. *J. Neuroinflammation* **12**: 1–13
- Keren-shaul H, Spinrad A, Weiner A, Colonna M, Schwartz M, Amit I, Keren-shaul H, Spinrad A, Weiner A, Matcovitch-natan O & Dvir-szternfeld R (2017) A Unique Microglia Type Associated with Restricting Development of Alzheimer ' s Disease Article A Unique Microglia Type Associated with Restricting Development of Alzheimer ' s Disease. *Cell*: 1–15
- Kummer MP, Vogl T, Axt D, Griep A, Vieira-saecker A, Jessen F, Gelpi E, Roth J & Heneka MT (2012) Mrp14 Deficiency Ameliorates Amyloid β Burden by Increasing Microglial Phagocytosis and Modulation of Amyloid Precursor Protein Processing. *J. Neurosci.* **32**: 17824–17829
- Liddel SA, Guttenplan KA, Clarke LE, Bennett FC, Bohlen CJ, Schirmer L, Bennett ML, Münch AE, Chung W-S, Peterson TC, Wilton DK, Frouin A, Napier BA, Panicker N, Kumar M, Buckwalter MS, Rowitch DH, Dawson VL, Dawson TM, Stevens B, et al (2017) Neurotoxic reactive astrocytes are induced by activated microglia. *Nature*
- Maier M, Peng Y, Jiang L, Seabrook TJ, Carroll MC & Lemere C a (2008) Complement C3 deficiency leads to accelerated amyloid beta plaque deposition and neurodegeneration and modulation of the microglia/macrophage phenotype in amyloid precursor protein transgenic mice. *J. Neurosci.* **28**: 6333–41
- Mendiola AS, Garza R, Cardona SM, Mythen SA, Lira SA, Akassoglou K & Cardona AE (2017) Fractalkine Signaling Attenuates Perivascular Clustering of Microglia and Fibrinogen Leakage during Systemic Inflammation in Mouse Models of Diabetic Retinopathy. *Front. Cell. Neurosci.* **10**: 1–15
- Olmos-Alonso A, Schettters STT, Sri S, Askew K, Mancuso R, Vargas-Caballero M, Holscher C, Perry VH & Gomez-Nicola D (2016) Pharmacological targeting of CSF1R inhibits microglial proliferation and prevents the progression of Alzheimer's-like pathology. *Brain* **139**: 891–907
- Perez OA, Yeung ST, Vera-Licona P, Romagnoli PA, Samji T, Ural BB, Maher L, Tanaka M & Khanna KM (2017) CD169⁺ macrophages orchestrate innate immune responses by regulating bacterial localization in the spleen. *Sci. Immunol.* **2**: eaah5520
- Rice RA, Spangenberg EE, Yamate-morgan H, Lee RJ, Arora RPS, Hernandez XMX, Tenner AJ, West BL & Green KN (2015) Elimination of Microglia Improves Functional Outcomes Following Extensive Neuronal Loss in the Hippocampus. *J. Neurosci.* **35**: 9977–9989

Thanks for sending us your revised manuscript. I asked referee #1 and 3 to re-review and their comments are provided below.

As you can see from these comments the referees appreciate the introduced changes and support publication here. There have a few remaining comments that I would like to ask you to respond to in a final revision. I think most of the issues can be resolved with text changes.

 REFEREE REPORTS:

Referee #1:

I think the findings are important. There are now several studies of systemic inflammations effects on neurodegenerative diseases and they take many different approaches. What the current study lacks in molecular detail it more than makes up for in quality and clarity of imaging to reach some useful conclusions about the contribution of NLRP3 to the effects of systemic LPS on microglial function and amyloid pathology.

For what it is worth, I believe that referee 2 is much too strong in stating that these do not constitute novel data and I agree with most of the author's responses to referee 2. I might add to the mechanistic account of how LPS effects on microglia could be mediated by NLRP3: microglial IL-1, which would be secreted due to NLRP3 processing (even if its expression levels are not altered, as is seen here) then has multiple effects on both microglial and astrocytes: IL-1 injected into the degenerating brain induces significant changes in microglial morphology, activation of astrocytes to synthesise excessive chemokines and significant infiltration of both monocytes and neutrophils (Hennessy et al., 2015, J Neurosci). It is not clearly known how microglia respond to systemic inflammation and that is something that needs to be resolved by the field as a whole and is not a specific weakness of the current study.

This is a revision and the rest of my comments refer to the revisions the authors have made in light of prior comments.

Figure 1 d,e. The data is no longer misrepresented but it certainly could be more simply written. The finding here is that branches, path length, maximum branch order are all reduced at 2 days but by 10 days they are once again indistinguishable from their baseline value. The authors text masks this simple observation with a lot of statistics-heavy text.

Figure 3. The authors have added further animals and the effects I noted in the prior figure, suggesting effects of LPS on amyloid beta even in NLRP3^{-/-} are no longer apparent. In general, the number of plaques per section is now lower in the AAP/NLRP3^{-/-} mice.

Figure 4. The following statement makes no sense: "Analysis by 2PLSM revealed that LPS did not lead to further morphological changes in microglia located in the vicinity of A β deposits (Fig 4a,b), which is defined as the microglia cells in a 60 μ m radius form the amyloid deposit core". It suggests that they have assessed morphology on the basis of the number of microglia within a certain radius. This is obviously a measure of microglial number, not morphology. Moreover, the green labeling is significantly more marked in the 2d and 10 d APP pictures than in the baseline picture. If that is not suggestive of altered morphology then what is?

With respect to the MX04 data: we pointed out that there was an apparent 2-way ANOVA interaction of strain and time post-LPS (this is important because it suggests that LPS might have OPPOSITE effects on phagocytosis of amyloid depending on the presence or absence of NLRP3). The authors have now performed that statistical analysis and confirm (in their responses) that this suggestion was correct. However they have not edited the text to reflect this (apparently ignoring the finding on the basis of post-hoc tests). My view is that the statistical interaction merits comment in the text.

Figure 6 CD169 can be expressed on both monocytic cells and on neutrophils. The pictures shown show no red labelling, suggesting complete overlap with IBA-1. Is this certainly the case or would a stronger amplification of the red channel show red cells that are not IBA1 positive?

Figure 7,8. Several changes made in the text here.

"In cortical areas, WT mice (5 and 15mo) had a higher proliferation rate as compared to Nlrp3^{-/-} animals (Fig 7)". I think it is important to state explicitly that LPS produces some proliferation at 2 days in WT and NLRP3 mice but that this is only NLRP3-dependent to a limited extent.

In figure 8, it is now corrected that LPS does indeed still produce proliferation in NLRP3^{-/-} mice.

Referee #3:

We appreciate the efforts the authors took to address our critiques. Except #5 all our points were dealt with adequately, in part by providing new experimental data, which improved the quality of the manuscript substantially. In consequence, only few minor issues need to be addressed:

1. Fig 4a and 5a: the assignment of the red and green signals is still not included in the figure and the figure legend
2. EV10: Please indicate which of the two bands in the Actin blot is the Actin band

2nd Revision - authors' response

30th Jun 2019

Referee #1

I think the findings are important. There are now several studies of systemic inflammations effects on neurodegenerative diseases and they take many different approaches. What the current study lacks in molecular detail it more than makes up for in quality and clarity of imaging to reach some useful conclusions about the contribution of NLRP3 to the effects of systemic LPS on microglial function and amyloid pathology.

For what it is worth, I believe that referee 2 is much too strong in stating that these do not constitute novel data and I agree with most of the author's responses to referee 2. I might add to the mechanistic account of how LPS effects on microglia could be mediated by NLRP3: microglial IL-1, which would be secreted due to NLRP3 processing (even if its expression levels are not altered, as is seen here) then has multiple effects on both microglial and astrocytes: IL-1 injected into the degenerating brain induces significant changes in microglial morphology, activation of astrocytes to synthesise excessive chemokines and significant infiltration of both monocytes and neutrophils (Hennessy et al., 2015, J Neurosci). It is not clearly known how microglia respond to systemic inflammation and that is something that needs to be resolved by the field as a whole and is not a specific weakness of the current study.

Response: we thank the reviewer for supporting our study.

This is a revision and the rest of my comments refer to the revisions the authors have made in light of prior comments.

Figure 1 d,e. The data is no longer misrepresented but it certainly could be more simply written. The finding here is that branches, path length, maximum branch order are all reduced at 2 days but by 10 days they are once again indistinguishable from their baseline value. The authors text masks this simple observation with a lot of statistics-heavy text.

Response: The reviewer is right in its observation, however we have to mention that despite not observing morphological differences between 10 days and baseline conditions for 15mo mice, we did not observe between 10 days and 2 days. We have simplified the sentence and removed the statistical text (Page 6 lines 140-143).

Figure 3. The authors have added further animals and the effects I noted in the prior figure, suggesting effects of LPS on amyloid beta even in NLRP3^{-/-} are no longer apparent. In general, the number of plaques per section is now lower in the AAP/NLRP3^{-/-} mice.

Response: we thank the reviewer.

Figure 4. The following statement makes no sense: "Analysis by 2PLSM revealed that LPS did not lead to further morphological changes in microglia located in the vicinity of A β deposits (Fig 4a,b), which is defined as the microglia cells in a 60 μ m radius form the amyloid deposit core". It suggests that they have assessed morphology on the basis of the number of microglia within a certain radius. This is obviously a measure of microglial number, not morphology. Moreover, the green labeling is significantly more marked in the 2d and 10 d APP pictures than in the baseline picture. If that is not suggestive of altered morphology then what is?

Response: We wanted to make clear what vicinity of the plaque means. Additionally, we reconstructed every individual cell for morphological quantification. The fact that every cell is reconstructed individually, circumvent any difference in the number (a difference we are not addressing in this figure). Regarding the difference in the intensity of the green fluorescent protein (GFP), the mice express GFP under the endogenous *Cx3-cr1* promotor. The reviewer is right in the observing that there is more GFP after LPS injection, this is mainly due to the fact that more cells are around the plaque. As mentioned above, the number of cells is not addressed in this figure and the morphological reconstruction of every individual cell circumvent any difference in number.

With respect to the MX04 data: we pointed out that there was an apparent 2-way ANOVA interaction of strain and time post-LPS (this is important because it suggests that LPS might have OPPOSITE effects on phagocytosis of amyloid depending on the presence or absence of NLRP3). The authors have now performed that statistical analysis and confirm (in their responses) that this suggestion was correct. However they have not edited the text to reflect this (apparently ignoring the finding on the basis of post-hoc tests). My view is that the statistical interaction merits comment in the text.

Response: we apologize for not having included the comment in the previous version. We have now included a comment on the nature of the statistical interaction (page 9 lines 236-239).

Figure 6 CD169 can be expressed on both monocytic cells and on neutrophils. The pictures shown show no red labelling, suggesting complete overlap with IBA-1. Is this certainly the case or would a stronger amplification of the red channel show red cells that are not IBA1 positive?

Response: we thank the reviewer for raising the concern. We have not seen any CD169 positive cell that is not Iba-1 positive. Regarding amplification of the red channel, we have taken every picture with the same look-up-tables settings.

Figure 7,8. Several changes made in the text here.

"In cortical areas, WT mice (5 and 15mo) had a higher proliferation rate as compared to Nlrp3^{-/-} animals (Fig 7)". I think it is important to state explicitly that LPS produces some proliferation at 2 days in WT and NLRP3 mice but that this is only NLRP3-dependent to a limited extent.

Response: We have rephrased the sentence in order to explicitly address the concern raised by the reviewer (page 12 lines 303-305).

In figure 8, it is now corrected that LPS does indeed still produce proliferation in NLRP3^{-/-} mice.

Response: we thank the reviewer for the suggestion

Referee #3

We appreciate the efforts the authors took to address our critiques. Except #5 all our points were dealt with adequately, in part by providing new experimental data, which improved the quality of the manuscript substantially. In consequence, only few minor issues need to be addressed:

1. Fig 4a and 5a: the assignment of the red and green signals is still not included in the in the figure and the figure legend

Response: Figures 4a and 5a are two-photon images. In figure 4a red channel represents dextran-red and the green channel represents the green fluorescent protein (GFP). In figure 5a green channel represents GFP. These corrections were included to figure 4a and 5a for the revised version of the manuscript.

2. EV10: Please indicate which of the two bands in the Actin blot is the Actin band

Response: we thank the reviewer for the comment. The actin band in the blot is represented by the upper band. Please be aware that for the revised version of the manuscript Fig EV10 is now Fig EV4.

3rd Editorial Decision

3rd Jul 2019

Thank you for sending me the revised manuscript I have now had a chance to take a look at everything and all looks good.

I am therefore very pleased to accept the manuscript for publication here.

Corresponding Author Name: Michael T. Heneka

Manuscript Number: EMBOJ-2018-101064R